# QAEncoder: Towards Aligned Representation Learning in Question Answering System

## Abstract

Modern QA systems entail retrieval-augmented generation (RAG) for accurate and trustworthy responses. However, the inherent gap between user queries and relevant documents hinders precise matching. Motivated by our conical distribution hypothesis, which posits that potential queries and documents form a cone-like structure in the embedding space, we introduce QAEncoder, a training-free approach to bridge this gap. Specifically, QAEncoder estimates the expectation of potential queries in the embedding space as a robust surrogate for the document embedding, and attaches document fingerprints to effectively distinguish these embeddings. Extensive experiments on fourteen embedding models across six languages and eight datasets validate QAEncoder's alignment capability, which offers a plug-and-play solution that seamlessly integrates with existing RAG architectures and training-based methods.

## 1 Introduction

> *"What I cannot create, I do not understand."* — Richard Feynman

Question Answering (QA) systems aim to generate accurate responses to user queries with applications in customer service (Xu et al., 2024), search engine (Ojokoh & Adebisi, 2018), healthcare (Guo et al., 2022) and education (Levonian et al., 2023), necessitating proficiencies in information retrieval, comprehension, and generation. Modern QA systems leverage large language models (LLMs) such as ChatGPT (Achiam et al., 2023), supplemented with retrieval-augmented generation (RAG) to address issues of outdated or hallucinatory information, especially for rapidly evolving knowledge bases (Lewis et al., 2020; Huang et al., 2023; Gupta et al., 2024). The efficacy of RAG hinges on its retrieval module for identifying relevant documents from a vast corpus. Dense retrievers (Lewis et al., 2020; Hofstätter et al., 2021), contrasted with keyword-matching-based sparse retrievers (Jones, 1973; Robertson & Zaragoza, 2009), have enabled efficient and precise retrieval by mapping queries and documents into a shared vector space. Despite advancements, a significant challenge that persists is bridging the semantic and syntactic gap between user queries and documents, known as *the document-query gap* (Zheng et al., 2020). Three main approaches have emerged to address this challenge: training-based alignment, document-centric alignment, and query-centric alignment.

Training-based approaches (Dong et al., 2022; Li et al., 2022; W et al., 2023; Zhang et al., 2024a; Khanna & Subedi, 2024) fine-tune or train embedding models from scratch with dedicated QA datasets to close the representation of relevant queries and documents, but struggle to fully generalize across new domains (Suprem & Pu, 2022). Furthermore, catastrophic forgetting necessitates updating embeddings for the previously encoded corpus when new data is learned (Pan et al., 2024), which is resource-intensive and practically unaffordable. Document-centric approaches (Wang et al., 2023b; Gao et al., 2023; Kim & Min, 2024) generate pseudo-documents for user queries by LLMs, which are then used to retrieve relevant information from the corpus, bridging the document-query gap by LLMs' zero-shot generalization capabilities. However, pseudo-document generation is both costly and time-consuming, and increases the risk of hallucinated information (Wang et al., 2023b). Contrastly, query-centric methods circumvent these problems.

Query-centric methods (Nogueira et al., 2019; Cheriton, 2019; Mallia et al., 2021b) generate and index document-relevant questions to align the indexed context with user queries. However, existing

solutions are predominantly based on sparse retrievers and have not fully leveraged the potential of dense retrievers. Combining query-centric methods with dense retrievers remains heavily under-explored. The naive solution can be storing all QA pairs into a vector database, but with several notable challenges:

C1. **Expanded Index Size.** Storing all QA pairs significantly increase the index size, leading to substantial expansion in storage requirements, especially problematic for large-scale corpora.

C2. **Prolonged Retrieval Times.** The index expansion also results in extended retrieval times. For dense retrievers, the expanded index size can result in linearly increased search time in both exhaustive and non-exhaustive search (Douze et al., 2024) and hurt the recall performance in non-exhaustive case (Zhao et al., 2023).

C3. **Limited Query Handling.** Although storing QA pairs individually can address predicted queries, this approach lacks robustness when confronted with the wide-ranging and diverse nature of potential queries. Such a method may be unable to effectively handle rephrasing in linguistic style, sentence structure, or even vocabulary, thereby compromising the overall reliability and performance of the system (Alting von Geusau & Bloem, 2020).

**Motivation** Inspired by Feynman's philosophy of learning (Reyes et al., 2021), we believe that effective information retrieval for QA systems extends beyond mere information storage and fundamentally depends on the active processing, involving the query formulation. This process mirrors human learning, where a deeper understanding of stored knowledge is achieved through thoughtful reflection and inquiry. For instance, the well-established 5W1H framework (Who, What, When, Where, Why, How) (Jinks, 2019) can be employed to systematically deconstruct information, fostering a comprehensive and nuanced understanding. To circumvent and solve the aforementioned challenges, we continue the research line of query-centric methods and introduce an innovative approach called *QAEncoder*. QAEncoder is motivated by a key observation termed *conical distribution hypothesis*. Specifically, for a given document, its potential queries are embedded approximately within a single cluster on some hyperplane in the semantic space, while the document embedding lies on the perpendicular line intersecting the cluster center. Hence, the projection of the document embedding onto this hyperplane, i.e. the cluster center, is significantly closer to the potential queries, and can be utilized to optimize the original document embedding.

As demonstrated in Fig. 1, our method initially generates diversified queries (e.g. 5W1H) and then estimates the cluster center by the Monte Carlo method. The similarity matrix reveals that, for any query, the mean-query similarity is significantly higher than both document-query and other query-query similarities. Hence, we advocate using the cluster center as a surrogate for the document embedding in QA systems, which bridges the document-query gap robustly without extra index size and retrieval latency.

Despite these advantages, the basic implementation encounters a critical challenge. While enhancing similarity with user queries, it simultaneously reduces the distinguishability between document representations, as they all become query-like. To address this side effect, we further introduce *document fingerprint strategies* to infuse unique document identities into representations and enable state-of-the-art performance.

**Contributions** The main contributions can be summarized as follows:

• **Methodological Innovations.** We pioneer to bridge the document-query gap in dense retrievers from the query-centric perspective. Our method, QAEncoder, not only avoids extra index storage, retrieval latency, training cost and hallucination, but also guarantees diversified query handling and robust generalization. We further propose document fingerprint strategies to address the side effect of distinguishability and achieve state-of-the-art performance.

• **Theoretical Discovery.** We formulate the conical distribution hypothesis, providing a theoretical foundation for the alignment of document and query embeddings, which is validated through extensive empirical analysis and provides deeper insights into semantic space in QA systems.

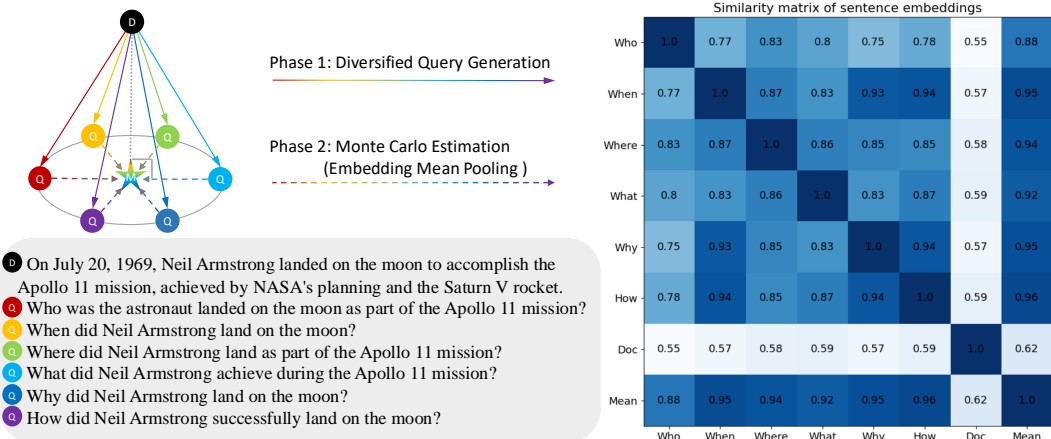

Figure 1: Illustration of QAEncoder's alignment process: Solid lines represent diversified query generation, while dashed lines indicate Monte Carlo estimation. The heatmap depicts the similarity scores among the embeddings of the different queries, the document, and the mean estimation.

- **Practical Applications.** QAEncoder is designed as a truly plug-and-play solution, seamlessly integrating with existing RAG architectures and training-based methods. This integration significantly enhances system performance with minimal modifications required.

## 2 RELATED WORKS

**Retrieval-augmented QA systems.** Retrieval-augmented generation significantly improves large language models in QA systems by incorporating a retrieval module that fetches relevant information from external knowledge sources (Févry et al., 2020; Guu et al., 2020; Izacard & Grave, 2021; Zhao et al., 2024). Retrieval models have evolved from early sparse retrievers, such as TF-IDF (Jones, 1973) and BM25 (Robertson & Zaragoza, 2009), which rely on word statistics and inverted indices, to dense retrieval strategies (Lewis et al., 2020) that utilize neural representations for enhanced semantic matching. Advanced methods, such as Self-RAG (Asai et al., 2023) which determines if additional information is required and evaluates the relevance of retrieved content, and RAG-end2end (Siriwardhana et al., 2023) that jointly trains the retriever and generator, represent significant developments in this area. However, these methods still ignore the inherent semantic gap between queries and documents.

**Training-based alignment.** Training-based approaches bridge the document-query gap generally by contrastive learning (Xiong et al.; Qu et al., 2021) or knowledge distillation (Zhang et al., 2024a; Khanna & Subedi, 2024). For instance, Dong et al. (2022) showed parameter sharing of the query encoder and the document encoder improves overall performance by projecting queries and documents into shared space. Dual-Cross-Encoder (Li et al., 2022) and Query-as-context (W et al., 2023) train embedding models from scratch with paired document-query samples. E5 (Wang et al., 2024a) and GTE (Zhang et al., 2024b) concatenate different prompts before queries for instruction-tuned embedding models that enhance downstream task adaptability, especially for QA systems. However, training-based methods face high cost, generalization difficulty (Suprem & Pu, 2022), and catastrophic forgetting issue (Pan et al., 2024). Catastrophic forgetting, where the encoder continually trained on new data loses previous function, is particularly detrimental as it necessitates the costly re-encoding of entire corpora. Especially note that, unsupervised domain adaptation methods like GPL (Wang et al., 2021), CAI (Iida & Okazaki, 2022) and AugTriever (Meng et al., 2022) are also of great importance due to the annotation-free nature. E.g., GPL and AugTriever use pseudo-queries as supervision for contrastive learning in new domains. However, they also face multi-domain adaptation challenges like task imbalance and catastrophic forgetting (Saunders, 2022).

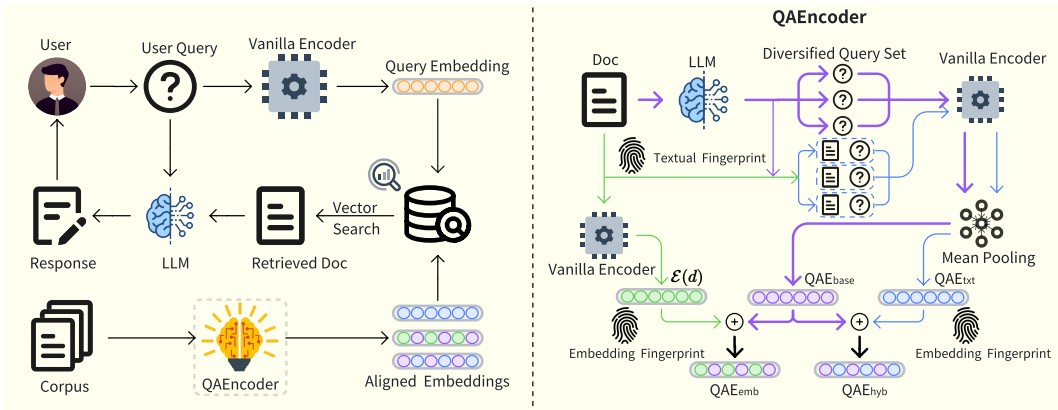

Figure 2: Architecture of QAEncoder. Left: Corpus documents are embedded using QAEncoder to obtain query-aligned representations for indexing. User queries are encoded with a vanilla encoder and used to retrieve relevant documents. Right: Internal mechanism of QAEncoder. QAEncoder addresses the document-query gap by generating a diverse set of queries for each document to create semantically aligned embeddings. Additionally, document fingerprint strategies are employed to ensure document distinguishability.

**Document-centric alignment.** Document-centric methods, such as HyDE (Gao et al., 2023) and Query2doc (Wang et al., 2023b), dynamically transform user queries into pseudo-documents using LLMs for both sparse and dense retrievers. QA-RAG (Kim & Min, 2024) advances by implementing a two-way retrieval mechanism that utilizes both user query and pseudo-documents for respective retrieval. However, their effectiveness is highly dependent on the quality of pseudo-documents generated, which are susceptible to hallucinations, especially for latest information. Furthermore, invoking LLMs for each query imposes substantial computational costs and increases latency, leading to degraded user experience.

**Query-centric alignment.** The seminal work, Doc2Query (Nogueira et al., 2019), mainly focuses on the vocabulary mismatch problem for sparse retrievers by expanding the document with keywords in predicted queries. The subsequent DocT5Query (Cheriton, 2019) improves by training a T5 model to predict queries. Based on DocT5Query, DeepImpact (Mallia et al., 2021a) further assigns weights on keywords with neural estimation for more precise sparse representations. Though these approaches have succeed for sparse retrievers, the integration with dense retrieval systems remains largely unexplored.

## 3 METHOD

### 3.1 PROBLEM FORMULATION

Given a query $q$ and a document corpus $\mathcal{D} = \{d_1, d_2, ..., d_i, ..., d_N\}$, our task is to retrieve a subset of $K$ most relevant documents $\mathcal{D}_+ = \{d_{i_1}, d_{i_2}, ..., d_{i_j}, ..., d_{i_K}\}$ through vector search. We define our embedding model as $\mathcal{E}(\cdot)$, which maps each document $d$ and query $q$ from the textual space $\mathcal{C}$ to a vector space $\mathbb{R}^r$. The semantic relevance is quantified by the cosine similarity between the query $q$ and each document $d$ in the embedding space, defined as:

$$sim(q, d) = \frac{\mathcal{E}(q)^T \mathcal{E}(d)}{\|\mathcal{E}(q)\|\|\mathcal{E}(d)\|}. \tag{1}$$

Furthermore, for each document $d$ in our dataset, we invoke the query generator $\mathcal{Q}(\cdot)$ multiple times to generate $n$ predicted queries $\{q_i\}_{i=1}^n$, where the cluster center in embedding space is captured by $\mathbb{E}[\mathcal{E}(\mathcal{Q}(d))]$ and estimated by Monte Carlo sampling $\overline{\mathcal{E}(\mathcal{Q}(d))}$.

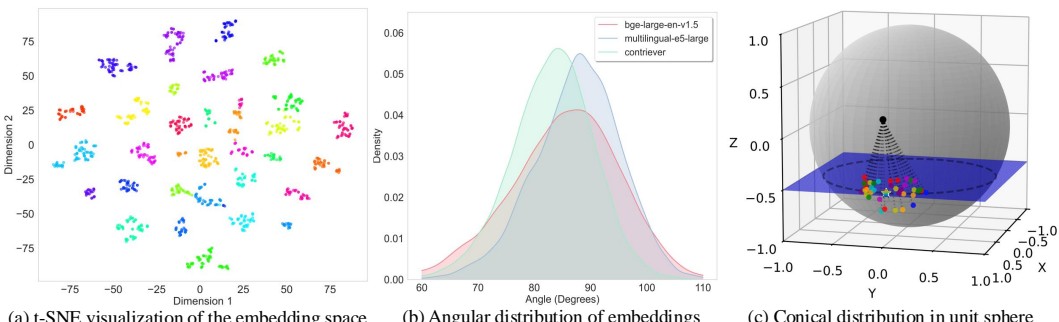

(a) t-SNE visualization of the embedding space  (b) Angular distribution of embeddings  (c) Conical distribution in unit sphere

Figure 3: Conical distribution hypothesis validation. The figure presents three visualizations supporting the conical distribution hypothesis: (a) t-SNE visualization of queries derived from various documents in the embedding space, illustrating distinct clustering behavior. (b) Angular distribution of document and query embeddings, showing the distribution of angles between $v_d = \mathcal{E}(d) - \mathbb{E}[\mathcal{E}(\mathcal{Q}(d))]$ and $v_{q_i} = \mathcal{E}(q_i) - \mathbb{E}[\mathcal{E}(\mathcal{Q}(d))]$. The angles form a bell curve just below 90°, supporting that $v_d$ is approximately orthogonal to each $v_{q_i}$ and serves as the normal vector. (c) 3D visualization illustrating the conical distribution of the document (black point) and query (colored points) embeddings within a unit sphere. The star indicates the queries' cluster center.

## 3.2 CONICAL DISTRIBUTION HYPOTHESIS

In this subsection, we formally define the highly simplified conical distribution hypothesis and validate its reasonableness with empirical analysis.

**Hypothesis 1** (Conical Distribution Hypothesis). *For any document d, the potential queries approximately form a single cluster on some hyperplane $\mathcal{H} = \{x \in \mathbb{R}^r \mid w \cdot x = b\}$ in the semantic space, where $w \in \mathbb{R}^r$ is the normal vector and $b \in \mathbb{R}$ is the bias term. Furthermore, the document embedding $\mathcal{E}(d)$ lies on the perpendicular line intersecting the cluster center $\mathbb{E}[\mathcal{E}(\mathcal{Q}(d))]$. Formally, the relationship can be represented as:*

$$\mathcal{E}(d) \approx \mathbb{E}[\mathcal{E}(\mathcal{Q}(d))] + \lambda w, \quad \lambda \in \mathbb{R}. \tag{2}$$

Fig. 3 presents the validation approach, and we leave the details in Appendix A.6. Furthermore, when the stronger hypothesis assuming the cluster follows the Gaussian distribution is adopted, more quantitative analysis results are derived in the appendix.

## 3.3 QAENCODER

Building upon the conical distribution hypothesis, we introduce a novel encoding method, $\text{QAE}_{\text{base}}$, which represents the document by the cluster center $\mathbb{E}[\mathcal{E}(\mathcal{Q}(d))]$ of potential queries, instead of the document embedding $\mathcal{E}(d)$ itself. Formally, the transformation is defined as follows:

$$\text{QAE}_{\text{base}}(d) = \mathbb{E}[\mathcal{E}(\mathcal{Q}(d))] \approx \overline{\mathcal{E}(\mathcal{Q}(d))} = \frac{1}{n}\sum_{i=1}^{n} \mathcal{E}(q_i). \tag{3}$$

However, the ideal embedding model should not only bring related entries closer in the embedding space but also separate unrelated entries as much as possible. Despite $\text{QAE}_{\text{base}}$ enhances document-query similarity compared to the original document embedding $\mathcal{E}(d)$, it poses the distinguishability issue.

### 3.3.1 DOCUMENT FINGERPRINT STRATEGIES

The distinguishability issue arises because incorporating too much query semantics into document representations suppresses their unique characteristics and renders unrelated documents more similar. To address this issue, we introduce the *document fingerprint strategies*, which involve three variations that enhance the uniqueness of document representations from different perspectives.

**Embedding fingerprint - QAE$_{\text{emb}}$.** The QAE$_{\text{emb}}$ strategy manipulates within the embedding space and reintroduces unique identity of the original document, i.e. the document embedding $\mathcal{E}(d)$. Specifically, QAE$_{emb}$ considers both the cluster center, QAE$_{\text{base}}$, and the document embedding, $\mathcal{E}(d)$, balancing their contributions using a hyperparameter $\alpha$. The adjusted embedding is formulated as follows:

$$\text{QAE}_{\text{emb}}(d) = (1 - \alpha) \cdot \mathcal{E}(d) + \alpha \cdot \text{QAE}_{\text{base}}(d) \approx (1 - \alpha) \cdot \mathcal{E}(d) + \alpha \cdot \frac{1}{n} \sum_{i=1}^{n} \mathcal{E}(q_i). \quad (4)$$

Due to the linearity of the inner product space, the cosine similarity, i.e. the inner product between the user query $q$ and QAE$_{\text{emb}}(d)$, is mathematically equivalent to the weighted sum of the similarities between the user query $q$ and both $\mathcal{E}(d)$ and QAE$_{\text{base}}(d)$, controlled by the hyperparameter $\alpha$.

**Textual fingerprint - QAE$_{\text{txt}}$.** The QAE$_{\text{txt}}$ strategy focuses on the textual space and injects the document identity in a more straightforward manner. Let us define the length of text $c$ in textual space as $|c|$. Before embedding, each document $d$ is enriched by concatenating it with predicted queries such that the ratio of the length of the predicted queries and the length of the original document is about $\beta$. Then, the final embedding is derived as the average representation of these enriched documents.

$$d_i^* = \text{concat}(d, \{q_j\}_{j=1}^{k}), \quad s.t. \ |\text{concat}(\{q_j\}_{j=1}^{k})| \approx \beta |d|.$$

$$\text{QAE}_{\text{txt}}(d) = \frac{1}{n} \sum_{i=1}^{n} \mathcal{E}(d_i^*) = \frac{1}{n} \sum_{i=1}^{n} \mathcal{E}(\text{concat}(d, \{q_j\}_{j=1}^{k})). \quad (5)$$

For each enriched document $d_i^*$, $\{q_j\}_{j=1}^{k}$ are randomly and independently sampled from the query generator $\mathcal{Q}(\cdot)$. We point out that the hyperparameter $\beta$ determines the proportion of the original document and the potential queries in the concatenated text. When $\beta$ is low, the concatenated text degrades to the original document; when $\beta$ is high, the document information is overwhelmed by excessive query content, and the resulting length mismatch also hinders semantic matching.

**Hybrid fingerprint - QAE$_{\text{hyb}}$.** The hybrid approach, QAE$_{\text{hyb}}$, seeks to combine the benefits of both the embedding and textual fingerprints for more sophisticated and nuanced document representations. Although QAE$_{\text{emb}}$ combines the document embedding $\mathcal{E}(d)$ and the cluster center $\mathbb{E}[\mathcal{E}(\mathcal{Q}(d))]$ through linear interpolation, inherent differences between these embeddings suggest that a simple linear interpolation should be suboptimal. Therefore, we explore the potential of substituting the document embedding $\mathcal{E}(d)$ in Equation 4 with QAE$_{\text{txt}}$, which fuses the semantics of both documents and queries.

$$\text{QAE}_{\text{hyb}}(d) = (1 - \alpha) \cdot \text{QAE}_{\text{txt}}(d) + \alpha \cdot \text{QAE}_{\text{base}}(d). \quad (6)$$

Note that a more straightforward alternative can be $\text{QAE}_{\text{hyb}'}(d) = (1 - \alpha) \cdot \mathcal{E}(d) + \alpha \cdot \text{QAE}_{\text{txt}}(d)$, which replaces QAE$_{\text{base}}$ in Equation 4 with QAE$_{\text{txt}}$ that integrates more document information. However, this substitution faces insufficient query semantics and limited improvements.

In our implementation, all the calculated embeddings are normalized for standardized cosine similarity measurement. Our experiments confirm QAE$_{\text{emb}}$ and QAE$_{\text{hyb}}$ outperform QAE$_{\text{base}}$ and QAE$_{\text{txt}}$.

## 4 EXPERIMENTS

**Datasets and Metrics.** To rigorously assess the effectiveness of QAEncoder, we employ six well-known datasets: Natural Questions (NQ) (Kwiatkowski et al., 2019), SQuAD (Rajpurkar et al., 2016), ELI5 (Fan et al., 2019), TriviaQA (Joshi et al., 2017), HotPotQA (Yang et al., 2018) and MSMARCO (Nguyen et al., 2016). These datasets are extensively used for evaluating machine reading comprehension and QA systems, providing a solid foundation for performance evaluation. However, due to classical datasets are frequently utilized for pre-training or fine-tuning embedding models[1], they fall short in objectively reflecting the generalized alignment capabilities for queries

---

[1]See the fine-tuning data of bge-m3 `https://huggingface.co/datasets/Shitao/bge-m3-data`.

and documents, particularly in rapidly evolving and updated knowledge base. Recognizing this limitation, we further test on two latest news datasets, the Chinese dataset CRUD-RAG (Lyu et al., 2024) and the multilingual dataset FIGNEWS (Zaghouani et al., 2024) covering English, Arabic, French, Hindi and Hebrew. Following the previous work (Li et al., 2022), we evaluate on the development subset; the evaluation metrics include MRR, MAP and NDCG, capturing both the recall and ranking capabilities. GPT-4o-mini serves as the query generator. We leave the results on MS-MARCO, TriviaQA, HotPotQA, FIGNEWS(French), FIGNEWS(Hindi), FIGNEWS(Hebrew) datasets and MAP metric in Appendix A.8 for space reason. We also leave implementation details in the appendix.

## 4.1 MAIN RESULTS

We mainly compare QAEncoder against the vanilla encoders. Query-centric methods for sparse retrievers are also presented. Our comparison involves the following approaches:

- **Sparse retrievers** - BM25 (Robertson & Zaragoza, 2009), Doc2Query (Cheriton, 2019) and DeepImpact (Mallia et al., 2021a).

- **Dense retrievers** - The state-of-the-art embedding models such as BGE models (Xiao et al., 2024) by BAAI, E5 models (Wang et al., 2023a) by Microsoft, GTE models (Zhang et al., 2024b) by Alibaba-NLP, Jina models (Günther et al., 2023) by Jina AI; Other well-known models like Contriever models (Izacard et al., 2021) by Facebook Research, BCEmbedding models (NetEase Youdao, 2023) by NetEase Youdao and the popular Text2Vec models (Xu, 2023). These models feature multilingual understanding and task-specific instruction tuning capabilities. DPR (Karpukhin et al., 2020), the seminal dense retriever, is also included. More detailed description can be found in Appendix A.3. We integrate them with QAEncoder to bridge the document-query gap. The search spaces for hyperparameters $\alpha$ and $\beta$ are [0.0, 0.15, 0.3, 0.45, 0.6, 0.75, 0.9] and [0.5,0.75,1.0,1.25,1.5] respectively. We adopt grid search for QAE$_{hyb}$.

### 4.1.1 EXPERIMENTS ON CLASSICAL DATASETS

As shown in Table 1, for sparse retrievers, Doc2Query exhibits some improvements over the standard BM25 method. However, the improvements are not as significant as DeepImpact. Conversely, the DeepImpact method, which builds on Doc2Query and incorporates BERT for term weight assignment, achieves better performance across all metrics and datasets. This not only suggests that non-neural retrievers have limited alignment capability, but also highlights the importance of weight adjustment of document and query information.

Dense retrievers generally outperform significantly compared with sparse retrievers. For the state-of-the-art embedding models such as BGE, E5, and GTE, integrating them with QAEncoder can result into considerable performance enhancements, particularly for rare or unseen datasets. For instance, the multilingual-e5-large model witnesses an MRR increase from 39.0 to 46.4 on the ELI5 dataset, due to the ELI5 dataset is heavily down-sampled during the fine-tuning (Wang et al., 2024a); while the gte-base-en-v1.5 model improves its MRR from 68.1 to 74.8 on the SQuAD dataset, due to the SQuAD dataset is not included in the fine-tuning data (Zhang et al., 2024b). It suggests that even state-of-the-art embedding models suffer limited generalization, while QAEncoder provides robust and generalized alignment. For other well-known models such as Contriever, BCEmbedding, and Text2Vec, QAEncoder significantly improves them due to their limited generalization caused by relatively smaller training datasets. E.g., the contriever model and its multilingual version, mcontriever, achieve improvements of 10.5 and 15.5 MRR points on the SQuAD dataset respectively. More data is available in the appendix.

### 4.1.2 EXPERIMENTS ON LATEST DATASETS

In scenarios such as search engine, financial analysis, and news QA, large volumes of new data constantly emerge and are indexed into retrieval base for accurate and up-to-date response. Hence, the alignment capability for previously unseen user queries and relevant documents is crucial for embedding models in RAG systems. We experiment on the latest news datasets, the multilingual dataset FIGNEWS and the Chinese dataset CRUD-RAG, to avoid data leakage and mimic the real-world scenarios.

Table 1: Retrieval performance on classical datasets NQ, SQuAD and ELI5. Higher is better, with the best one bolded. Hyperparameters including QAEncoder variants and weight terms $\alpha$, $\beta$ are optimized simultaneously for all classical datasets. '-'denotes default or null values.

| Model | Param | NQ | | SQuAD | | ELI5 | |
|---|---|---|---|---|---|---|---|
| | | MRR@8 | NDCG@8 | MRR@8 | NDCG@8 | MRR@8 | NDCG@8 |
| **Sparse** | | | | | | | |
| BM25 | - | 17.9 | 12.8 | 46.2 | 48.2 | 12.6 | 11.5 |
| Doc2Query | - | 18.2 | 13.6 | 45.5 | 48.2 | 12.6 | 11.8 |
| DeepImpact | - | **21.8** | **15.7** | **47.8** | **50.0** | **12.9** | **11.9** |
| **Dense** | | | | | | | |
| bge-large-en-v1.5 | - | 87.0 | 68.0 | 76.2 | 79.9 | 57.7 | 55.8 |
| | QAE$_{hyb}$, $\alpha = 0.15$, $\beta = 0.5$ | **88.2** | **69.1** | **78.2** | **81.8** | **59.1** | **57.3** |
| multilingual-e5-large | - | 86.0 | 66.3 | **86.2** | **88.9** | 39.0 | 37.6 |
| | QAE$_{hyb}$, $\alpha = 0.15$, $\beta = 1.5$ | **86.1** | **66.7** | 84.9 | 87.9 | **46.4** | **43.3** |
| gte-base-en-v1.5 | - | **86.2** | **67.6** | 68.1 | 72.4 | 54.5 | 51.6 |
| | QAE$_{hyb}$, $\alpha = 0.3$, $\beta = 0.5$ | 85.5 | 67.4 | **74.8** | **78.7** | **57.0** | **55.1** |
| jina-embeddings-v2-small-en | - | 82.4 | 63.3 | 69.5 | 73.6 | **54.3** | 51.6 |
| | QAE$_{hyb}$, $\alpha = 0.15$, $\beta = 0.5$ | **83.3** | **64.1** | **72.4** | **76.2** | 53.8 | 51.6 |
| contriever | - | 78.8 | 60.8 | 64.8 | 69.4 | 51.3 | 49.8 |
| | QAE$_{emb}$, $\alpha = 0.45$ | **84.0** | **65.7** | **74.9** | **78.9** | **55.7** | **54.4** |
| mcontriever | - | 52.1 | 37.6 | 49.3 | 54.9 | 43.0 | 40.6 |
| | QAE$_{hyb}$, $\alpha = 0.45$, $\beta = 0.75$ | **61.4** | **45.9** | **64.7** | **69.5** | **51.0** | **48.3** |
| bce-embedding-base-v1 | - | 74.4 | 55.5 | **77.2** | **81.3** | 47.3 | 45.0 |
| | QAE$_{emb}$, $\alpha = 0.3$ | **76.4** | **56.8** | 77.1 | 81.1 | **50.1** | **48.4** |
| text2vec-base-multilingual | - | 53.8 | 36.6 | 40.9 | 45.6 | 38.2 | 34.7 |
| | QAE$_{hyb}$, $\alpha = 0.6$, $\beta = 0.5$ | **66.6** | **47.2** | **56.2** | **60.6** | **43.4** | **40.8** |
| dpr-multiset-base | - | 77.3 | 60.3 | 59.3 | 64.9 | 59.6 | 57.3 |
| | QAE$_{emb}$, $\alpha = 0.45$ | **82.2** | **64.2** | **64.2** | **69.1** | **60.1** | **58.3** |
| dpr-single-nq-base | - | 77.6 | 60.7 | 60.3 | 66.1 | 57.3 | 55.4 |
| | QAE$_{emb}$, $\alpha = 0.45$ | **81.8** | **63.3** | **66.4** | **70.8** | **58.8** | **57.2** |

Table 2: Retrieval performance on the latest datasets FIGNEWS and CRUD-RAG. Higher is better, with the best one bolded. Hyperparameters including QAEncoder variants and weight terms $\alpha$, $\beta$ are optimized simultaneously for all lastest datasets. '-' denotes default or null values.

| Model | Param | FIGNEWS(English) | | FIGNEWS(Arabic) | | CRUD-RAG(Chinese) | |
|---|---|---|---|---|---|---|---|
| | | MRR@8 | NDCG@8 | MRR@8 | NDCG@8 | MRR@8 | NDCG@8 |
| bge-m3 | - | 74.3 | 78.2 | 77.7 | 80.5 | 86.8 | 88.9 |
| | QAE$_{txt}$, $\beta = 1.5$ | **77.1** | **80.8** | **80.0** | **82.8** | **89.0** | **90.8** |
| multilingual-e5-small | - | 70.9 | 74.7 | 74.0 | 77.1 | 81.7 | 84.2 |
| | QAE$_{hyb}$, $\alpha = 0.3$, $\beta = 0.5$ | **74.4** | **78.2** | **78.8** | **81.2** | **87.4** | **89.2** |
| multilingual-e5-base | - | 74.7 | 77.9 | 72.2 | 75.7 | 86.1 | 88.1 |
| | QAE$_{emb}$, $\alpha = 0.3$ | **77.4** | **80.7** | **77.1** | **80.0** | **88.5** | **90.5** |
| multilingual-e5-large | - | 73.8 | 77.5 | 76.6 | 79.9 | 85.7 | 88.1 |
| | QAE$_{hyb}$, $\alpha = 0.15$, $\beta = 1.25$ | **77.0** | **80.4** | **82.1** | **84.9** | **89.1** | **91.1** |
| gte-multilingual-base | - | 65.3 | 69.9 | 73.2 | 76.2 | 82.9 | 85.6 |
| | QAE$_{hyb}$, $\alpha = 0.15$, $\beta = 1.5$ | **75.3** | **79.0** | **76.2** | **79.0** | **85.5** | **88.0** |
| mcontriever | - | 32.8 | 36.3 | 40.1 | 44.2 | 71.7 | 75.8 |
| | QAE$_{hyb}$, $\alpha = 0.45$, $\beta = 1.25$ | **61.2** | **65.4** | **68.2** | **71.9** | **88.7** | **90.5** |
| bce-embedding-base-v1 | - | 58.9 | 63.3 | - | - | 76.8 | 80.3 |
| | QAE$_{hyb}$, $\alpha = 0.3$, $\beta = 0.5$ | **66.6** | **70.6** | - | - | **85.9** | **88.1** |
| text2vec-base-multilingual | - | 38.5 | 43.0 | 27.7 | 31.4 | 12.1 | 13.7 |
| | QAE$_{emb}$, $\alpha = 0.75$ | **55.2** | **59.3** | **51.3** | **54.8** | **55.3** | **58.6** |

As illustrated in Table 2, for the latest datasets, QAEncoder significantly enhances the alignment of user queries with relevant documents across both state-of-the-art embedding models and other well-known models. For instance, the gte-multilingual-base model's MRR metric on the FIGNEWS(English) dataset increases from 65.3 to 75.3. Similarly, the mcontriever model's MRR on the FIGNEWS(English) dataset improves from 32.8 to 61.2. Besides, the text2vec-base-multilingual model's MRR on the CRUD-RAG dataset rises from 12.1 to 55.3. These results con-

Table 3: Performance comparison of QAEncoder variants on the latest datasets FIGNEWS and CRUD-RAG. Higher is better, with the best one bolded. Hyperparameters are optimized simultaneously for all the latest datasets. $n$ indicates the number of predicted queries in $QA_{naive}$.

| Model | Param | FIGNEWS(English) | | FIGNEWS(Arabic) | | CRUD-RAG(Chinese) | |
|---|---|---|---|---|---|---|---|
| | | MRR@8 | NDCG@8 | MRR@8 | NDCG@8 | MRR@8 | NDCG@8 |
| bge-m3 | $QAE_{emb}, \alpha = 0.3$ | 76.2 | 80.0 | 80.0 | 82.6 | 88.7 | 90.6 |
| | $QAE_{txt}, \beta = 1.5$ | 77.1 | **80.8** | 80.1 | 82.8 | 89.0 | 90.8 |
| | $QAE_{hyb}, \alpha = 0.15, \beta = 1.5$ | **77.3** | 80.7 | **80.6** | **83.3** | **89.4** | **91.2** |
| | $QAE_{naive}, n = 10$ | 76.8 | 79.5 | 76.9 | 79.2 | 85.9 | 87.8 |
| multilingual-e5-large | $QAE_{emb}, \alpha = 0.45$ | **77.8** | **81.1** | 79.7 | 82.7 | **89.7** | **91.4** |
| | $QAE_{txt}, \beta = 1.5$ | 75.6 | 79.0 | 80.7 | 83.7 | 88.3 | 90.4 |
| | $QAE_{hyb}, \alpha = 0.15, \beta = 1.25$ | 77.0 | 80.5 | **82.1** | **84.9** | 89.1 | 91.1 |
| | $QAE_{naive}, n = 10$ | 76.8 | 79.4 | 76.5 | 79.2 | 84.9 | 86.8 |

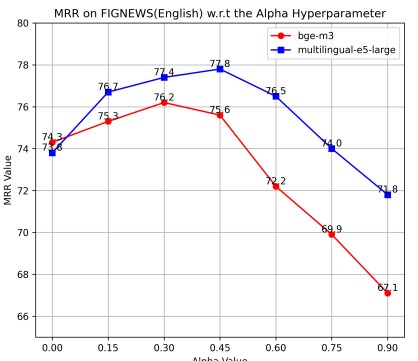
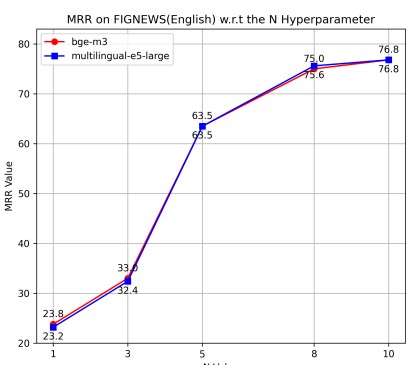

Figure 4: Ablation on $\alpha$ and $n$ hyperparameters for $QAE_{emb}$ and $QAE_{naive}$ on FIGNEWS(English) dataset. Left: The graph illustrates the impact of varying $\alpha$ values for $QAE_{emb}$, where MRR increases initially, peaks at $\alpha \approx 0.45$, and subsequently declines as $\alpha$ continues to rise. Right: The graph depicts the effect of varying the number of predicted queries for $QAE_{naive}$, with MRR improving as $n$ increases, approaching stability at $n = 10$. The curves for different models are mostly identical as the matching are largely driven by the alignment with the predicted individual queries.

firm QAEncoder's remarkably generalized alignment capability across various embedding models and multilingual datasets.

## 4.2 ANALYSIS AND DISCUSSION

For a more comprehensive assessment, we also report results of various QAEncoder ablations, i.e. $QAE_{emb}$, $QAE_{txt}$, $QAE_{hyb}$ and $QAE_{naive}$ which directly stores predicted queries. Finally, we also discuss the relationship between QAEncoder and training-based as well as document-centric methods.

### 4.2.1 ABLATION STUDIES

We present the performance comparison of QAEncoder variants on two state-of-the-art multilingual embedding models in Table 3. Generally, $QAE_{hyb}$ and $QAE_{emb}$ outperform the $QAE_{txt}$ and $QAE_{naive}$ approaches. For instance, for the bge-m3 model, $QAE_{hyb}$ consistently outperforms other variants. Conversely, the multilingual-e5-large model performs best with $QAE_{emb}$. However, the best performance differences between $QAE_{emb}$, $QAE_{txt}$, and $QAE_{hyb}$ are not substantial, demonstrating the robustness of our approach to hyperparameter variations. Regarding $QAE_{naive}$, it evidently underperforms other ablations, despite storing 10 times the number of embedding vectors. This leads to unacceptable storage management overhead and recall latency in large-scale production systems. We provide more granular ablation experiments in Fig. 4, and the convergence speed of Monte Carlo in Fig. 6.

Table 4: Performance comparison of QAEncoder with training-based and document-centric methods on the latest datasets FIGNEWS and CRUD-RAG. Higher is better, with the best one bolded. Hyperparameters are optimized simultaneously for all latest datasets. $n$ denotes the number of pseudo-documents in HyDE. '-' indicates default parameters.

| Model | Param | FIGNEWS(English) | | FIGNEWS(Arabic) | | CRUD-RAG(Chinese) | |
|---|---|---|---|---|---|---|---|
| | | MRR@8 | NDCG@8 | MRR@8 | NDCG@8 | MRR@8 | NDCG@8 |
| mcontriever-msmarco | - | 65.9 | 70.0 | 70.1 | 73.5 | 85.0 | 87.4 |
| | QAE$_{hyb}$, $\alpha = 0.3$, $\beta = 0.75$ | **72.1** | **76.3** | **77.2** | **80.2** | **88.6** | **90.6** |
| multilingual-e5-large-instruct | - | 66.7 | 70.7 | 74.9 | 78.1 | 79.8 | 82.5 |
| | QAE$_{hyb}$, $\alpha = 0.15$, $\beta = 1.5$ | **75.5** | **79.4** | **80.7** | **83.5** | **88.9** | **90.6** |
| mcontriever | - | **32.8** | **36.3** | **40.1** | **44.2** | **71.7** | **75.8** |
| | HyDE, $n = 8$ | 24.9 | 27.6 | 35.1 | 40.4 | 70.7 | 74.2 |
| multilingual-e5-large | - | **73.8** | **77.5** | **76.6** | **79.9** | **85.7** | **88.2** |
| | HyDE, $n = 8$ | 63.4 | 67.7 | 68.2 | 73.8 | 81.5 | 83.9 |

### 4.2.2 TRAINING-BASED AND DOCUMENT-CENTRIC METHODS

Training-based approaches mainly include two categories, fine-tuning on QA datasets and multi-task instruction datasets. We choose mcontriever-msmarco and multilingual-e5-large-instruct as representative models respectively. Finally, we consider HyDE (Gao et al., 2023), a well-known document-centric method for comparison.

Training-based and query-centric methods operate at training time and indexing time, respectively. Therefore, integrating these approaches cloud lead to more improvements. As illustrated in Table 4, both types of fine-tuned models significantly benefit from the QAEncoder. For instance, the mcontriever-msmarco model improves MRR from 70.1 to 77.2 on FIGNEWS(Arabic); the multilingual-e5-large-instruct model's MRR increases 8.8 and 9.1 MRR points on the FIGNEWS(English) and CRUD-RAG(Chinese) datasets, respectively.

For document-centric methods, we investigate the widely-reported hallucination phenomenon on latest datasets (Wang et al., 2023b; Gao et al., 2023; Kim & Min, 2024). Table 4 shows the recall performance heavily decreases for all the latest datasets and both models, attributed to the hallucination of pseudo-documents. Besides, the LLM invocation for pseudo-documents is both time-consuming and costly. In our case, the time for single LLM invocation is more than 2000ms while the time for vector search is less than 10ms. These highlight the irreplaceable importance and practicality of QAEncoder method.

## 5 CONCLUSION AND FUTURE WORK

In this paper, we introduce QAEncoder, a training-free approach to bridge the document-query gap for more advanced QA systems. Based on our conical distribution hypothesis and document fingerprint strategies, QAEncoder substitutes document embeddings with the expectation of query embeddings, enriched with document information. Notably, QAEncoder operates with zero additional index storage, retrieval latency, training costs, or risk of hallucination. Extensive experiments across multiple datasets, languages and embedding models further confirm its robust generalization, diversified query handling and compatibility with existing RAG architectures and training-based methods. **Future Work:** Despite these benefits, the current single-cluster hypothesis is overly simplistic and limits the performance improvement. The multi-cluster version such as Gaussian mixture models should be explored. Besides, since the query generator mainly generates simple queries, out-of-domain issues with complex, multi-hop queries could happen. Various strategies including data-driven proposal can be further investigated. We leave these research problems in the future work.

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

# A APPENDIX

## CONTENTS

## A.1   DATASET DETAILS

- MS-MARCO is a large-scale dataset specifically designed for machine reading comprehension. It comprises real user queries collected through Bing search, paired with corresponding passages retrieved from a comprehensive web document collection.

- Natural Questions (NQ) is a widely used dataset comprising real user queries submitted to the Google search engine, paired with relevant Wikipedia pages. Each query is annotated with a long answer (typically a paragraph) and a short answer (one or more entities). Each question-answer pair corresponds to a single Wikipedia page, ensuring clarity in evaluating information retrieval performance.

- TriviaQA is a dataset of question-answer-evidence triples, with over 650K entries, including 95K question-answer pairs created by trivia enthusiasts. The evidence documents, typically sourced from Wikipedia or other web sources, average six per question.

- HotPotQA contains 113K Wikipedia-based question-answer pairs that require multi-hop reasoning across multiple pages. Each pair includes a set of supporting sentences, which are treated as evidence documents.

- ELI5 is a question-answering dataset consisting of question-answer-evidence triples, where the questions are complex and often require detailed, multi-sentence responses. It contains 270K diverse and intricate questions that necessitate explanatory answers. To support each response, web search results are utilized as evidence documents, enhancing the reliability of the provided answers.

- SQuAD is a widely used reading comprehension dataset comprising 107K question-answer pairs derived from 536 Wikipedia articles. Each question are specific text segments from the relevant articles. In this dataset, the Wikipedia articles function as evidence documents, supplying the essential context required for accurate response retrieval.

- CRUD-RAG is a benchmark specifically designed for evaluating RAG systems, incorporating the latest high-quality news data that were not included in the training phase of the language models. It comprises more than 80K news articles sourced from prominent Chinese news websites, all published after July 2023. From the set of queries generated by GPT-4o-mini for each document, we randomly sample one to serve as the test query. The original news is designated as the evidence documents for recall evaluation, ensuring queries are associated with exactly relevant documents.[2]

- FIGNEWS is a multilingual news post dataset designed to examine bias and propaganda within news articles across different languages. It consists of 15,000 publicly available news posts collected from verified blue-check accounts between October 7, 2023, and January 31, 2024. The dataset includes posts in five languages—English, Arabic, Hebrew, French, and Hindi—distributed evenly across 15 batches, each containing 1,000 posts. Each batch consists of 200 posts for each language. Similar to CRUD-RAG, we randomly sample one predicted query generated by GPT-4o-mini for each document and use the original news as the evidence documents for recall evaluation.

---

[2]LlamaIndex adopts the same common practice and provides templated workflow, i.e. generating queries for recall and rerank evaluation. See https://www.llamaindex.ai/blog/boosting-rag-picking-the-best-embedding-reranker-models-42d079022e83.

## A.2 METRIC DETAILS

For SQuAD, CRUD-RAG and FIGNEWS benchmarks, MAP metric is equivalent to MRR, as there is only one related document per query.

- Mean Reciprocal Rank (MRR): Mean Reciprocal Rank: is a statistic measure used to evaluate the effectiveness of a retrieval system by calculating the reciprocal of the rank at which the first relevant result appears. The mathematical formulation is:

$$\text{MRR} = \frac{1}{|Q|} \sum_{i=1}^{|Q|} \frac{1}{\text{rank}_i}$$

Where: - $|Q|$ is the number of queries. - $\text{rank}_i$ is the rank position of the first relevant document for the $i$-th query.

- Mean Average Precision (MAP): Mean Average Precision is a measure that combines precision and recall. It computes the average precision value for a set of queries and then averages these values. The mathematical formulation is:

$$\text{MAP} = \frac{1}{|Q|} \sum_{i=1}^{|Q|} \text{AP}(i),$$

Where: - $|Q|$ is the number of queries. - $\text{AP}(i)$ is the average precision for the $i$-th query. The average precision for a single query is given by:

$$\text{AP} = \frac{1}{R} \sum_{k=1}^{n} P(k) \cdot \text{rel}(k),$$

Where: - $R$ is the total number of relevant documents for the query. - $n$ is the number of retrieved documents. - $P(k)$ is the precision at cut-off $k$. - $\text{rel}(k)$ is an indicator function equating to 1 if the document at rank $k$ is relevant, otherwise 0.

- Normalized Discounted Cumulative Gain (NDCG): Normalized Discounted Cumulative Gain is a measure of ranking quality that takes into account the positions of the relevant documents. It is based on the concept of discounting the relevance of documents based on their position in the result list. The mathematical formulation is:

$$\text{NDCG} = \frac{DCG_p}{IDCG_p},$$

Where: - $DCG_p$ is the Discounted Cumulative Gain at position $p$. - $IDCG_p$ is the Ideal Discounted Cumulative Gain at position $p$, which is the DCG score of the perfect ranking. The Discounted Cumulative Gain at position $p$ is given by:

$$DCG_p = \sum_{i=1}^{p} \frac{2^{\text{rel}_i} - 1}{\log_2(i + 1)},$$

Where: - $\text{rel}_i$ is the relevance score of the document at rank $i$.

The Ideal Discounted Cumulative Gain $IDCG_p$ is computed in the same way as $DCG_p$, except that the documents are ideally sorted by relevance.

## A.3 BASELINE DETAILS

- BM25 (Robertson & Zaragoza, 2009) is the traditional lexical retriever based on term relevance and frequency, regarded as the most popular variation of TF-IDF.
- DocT5Query (Cheriton, 2019) appends generated queries to the document before building the inverted index of BM25.
- DeepImpact (Mallia et al., 2021a) leverages DocT5Query to enrich the document collection and estimates the semantic importance of tokens with fine-tuned BERT.

- BGE models (Xiao et al., 2024)by BAAI, Jina models (Günther et al., 2023) by Jina AI, E5 models (Wang et al., 2023a) by Microsoft, GTE models (Zhang et al., 2024b) by Alibaba-NLP are the most advanced embedding models, featuring multilingual understanding and task-specific instruction tuning capabilities. We choose both vanilla encoders and QA-specific instruction-tuned encoders for test.

- Contriever models (Izacard et al., 2021) are developed by Facebook Research, including contriever, mcontriever, contriever-msmacro and mcontriever-msmacro. mcontriever serves as the multilingual version of contriever. contriever-msmacro and mcontriever-msmacro are further fine-tuned on the MS-MACRO dataset for bridging the document-query gap.

- BCEmbedding models (NetEase Youdao, 2023), developed by NetEase Youdao, are bilingual and crosslingual embedding models in English and Chinese. BCEmbedding serves as the cornerstone of Youdao's RAG-based QA system, QAnything, an open-source project widely integrated in commercial products like Youdao Speed Reading and Youdao Translation. We choose bce-embedding-base for test.

- Text2Vec models (Xu, 2023) is a popular open-source project that implements Word2Vec (Mikolov et al., 2013), RankBM25, BERT (Devlin et al., 2019), Sentence-BERT (Reimers & Gurevych, 2019), CoSENT and other text representation models. We test its most prominent model, text2vec-base-multilingual, which supports multiple languages, including German, English, Spanish, French, Italian, Dutch, Polish, Portuguese, Russian, and Chinese.

- DPR models (Karpukhin et al., 2020) by Facebook adopt a bi-encoder architecture. DPR models fine-tuned BERT on pairs of questions and passages without additional pretraining, achieving superior performance compared to traditional methods like BM25. There are two main variants, dpr-multiset-base and dpr-single-nq-base. dpr-multiset-base fine-tuned on Natural Questions (NQ), TriviaQA, WebQuestions (WQ), and CuratedTREC (TREC), while dpr-single-nq-base fine-tuned on the NQ dataset only.

- Training-based approaches mainly include two categories, fine-tuning on QA datasets or multi-task instruction datasets. For fine-tuning on QA datasets, we choose mcontriever-msmarco for test, an enhanced variant of the mcontriever model that has been fine-tuned on the MS-MARCO. The second category involves fine-tuning models on multi-task instruction datasets, where distinct prompt prefixes are appended to the input text, enabling the model to effectively differentiate between various tasks. In this category, we test the multilingual-e5-large-instruct (Wang et al., 2024a) developed by Microsoft, which leverages synthetic instruction data (Wang et al., 2024b) for fine-tuning.

- Document-centric methods instruct LLMs to generate a pseudo-document for each query. The pseudo-document aims to capture relevant information but does not correspond to a real document and may contain inaccuracies and hallucinations. Subsequently,the pseudo-document is encoded, and its embedding is utilized to retrieve similar real documents based on vector similarity. We choose HyDE (Gao et al., 2023) for test, which generates multiple pseudo-documents and fuses their embeddings by mean pooling for retrieval.

### A.4 PROMPTS

We employ specialized prompts to instruct gpt-4o-mini as the question and pseudo-document generator. Only the English version is presented due to LaTeX compilation issues with non-English languages.

```
Question Generator Prompt

Context information is below.
---------------------
[Document]
---------------------
Given the context information and not prior knowledge, generate
    only questions based on the below query.
You are a Teacher/Professor. Your task is to setup [Number of
    Questions] questions for an upcoming quiz/examination. The
    questions should be diverse in nature across the document.
    Restrict the questions to the information provided, and avoid
    ambiguous references.

Output Format:
```json
[
    "1. question",
    "2. question",
    ...
]
```
```

```
Pseudo-Document Generator Prompt

Please write a passage to answer the question.
Question: [Question]
Output Format:
```json
{
    "passage": ""
}
```
```

## A.5 HYPERPARAMETER SEARCH AND SELECTION SUGGESTION

$QAE_{emb}$ maintains competitive performance with single hyperparameter. Hence, $QAE_{emb}$ is recommended for accelerating HP search.

Firstly, we believe that the optimal hyperparameters are primarily influenced by the inherent characteristics of the embedding model, i.e. the geometric property of embedding space. Therefore, a one-turn search should be sufficient for a given embedding model. That's why we optimize hyperparameters simultaneously across multiple datasets.

Secondly, the one-turn search can also be accelerated under our framework. Indeed, as Fig. 4 shows, the performance of $QAE_{emb}$ empirically follows a consistent trend across various models and datasets: it initially rises and then falls as $\alpha$ increases, peaking between 0.3 and 0.6. This unimodal phenomenon enables ternary search with logarithmic trails rather than brute-force search.

Finally, the property of datasets also slightly influences the optimal hyperparameters. Specifically, the optimal $\alpha$ for classical datasets is marginally lower than that for latest datasets (refer to Tables 5 and 6 for details). Therefore, selecting the optimal $\alpha$ based on classical datasets represents a cautious and robust strategy, ensuring consistent improvement across both classical and latest datasets.

## A.6 PROOF OF THE CONICAL DISTRIBUTION HYPOTHESIS

This subsection provides the proof of the Conical Distribution Hypothesis, which proposes that potential queries form a distinct cluster on a hyperplane in semantic space.

*Proof.* Our validation is structured from three core aspects:

- **Single-cluster sub-hypothesis verification.** As illustrated in Fig. 3(a), we validate the single-cluster sub-hypothesis by visualizing the embedding space using t-SNE dimensionality reduction techniques. This visualization displays that the predicted queries for each document form distinct and cohesive clusters (different colored). And these clusters are notably distant from the clusters of other documents, thereby supporting the single-cluster sub-hypothesis.

- **Perpendicular sub-hypothesis verification.** To further assess the perpendicular sub-hypothesis, let $v_d = \mathcal{E}(d) - \mathbb{E}[\mathcal{E}(\mathcal{Q}(d))]$ and $v_{q_i} = \mathcal{E}(q_i) - \mathbb{E}[\mathcal{E}(\mathcal{Q}(d))]$ be the vectors from the cluster center to the document embedding and the individual query embedding, respectively. As illustrated in Fig. 3(b), the degree distribution between vector $v_d$ and vector $v_{q_i}$ exhibits a bell-shaped curve. The mean value is slightly less than 90 degrees, and the primary range of distribution lies between 75 and 100 degrees, which confirms that $v_d$ is approximately orthogonal to each $v_{q_i}$ and can be regarded as the normal vector to some hyperplane $\mathcal{H}$.

- **Conical distribution in unit sphere demonstration.** Finally, we illustrate the highly simplified conical distribution hypothesis within the unit sphere embedding space, as most embedding models utilize normalized embedding vectors. As depicted in Fig. 3(c), the embeddings of potential queries form a cluster on the surface of the unit sphere, with each point color-coded. The center of the cluster is indicated by a star, while the document embedding is represented by a black point positioned above the cluster. It is evident that these elements form a distorted cone, aligning with the above hypothesis and the degree distribution experiment.

$\square$

### A.7 STRONG CONICAL DISTRIBUTION HYPOTHESIS

In this subsection, we further substantiate the original hypothesis that the potential queries adhere to a Gaussian distribution: For any document $d$, the potential queries in the embedding space approximately follow a Gaussian distribution, characterized by a mean $\mu$ and covariance matrix $\Sigma$. Refer to Appendix A.7.3 for detailed validation.

### A.7.1 MAIN THEOREM

Building on this Gaussian assumption, we derive bounds on the cosine similarity between potential query embeddings and both the document embedding and the mean vector.

**Theorem 1.** *(Concentration Inequalities for Cosine Similarities in Embedding Spaces) Let $\mathbf{q} \sim \mathcal{N}(\mu, \Sigma)$ denote a random vector representing the distribution of potential queries of document $d$ in the unit sphere embedding space, where $\mu \in \mathbb{R}^r$ is the mean vector and $\Sigma \in \mathbb{R}^{r \times r}$ is the covariance matrix. Let $\mathbf{d} \in \mathbb{R}^r$ be the embedding of document $d$, and let $\theta$ be the angle between $\mu$ and $\mathbf{d}$ such that $\cos(\theta) = \mu^\top \mathbf{d}$. Assume that both $\mu$ and $\mathbf{d}$ are unit vectors. Then, the following properties hold:*

*1. The concentration inequality for the cosine similarity measure of $\mathbf{q}$ with $\mathbf{d}$ :*

$$\mathbb{P}\left(\left|\mathbf{q}^\top \mathbf{d} - \cos(\theta)\right| \geq t\right) \leq 2 \exp\left(-\frac{t^2}{2\mathbf{d}^\top \Sigma \mathbf{d}}\right). \tag{7}$$

*2. The concentration inequality for the cosine similarity measure of $\mathbf{q}$ with $\mu$ :*

$$\mathbb{P}\left(\left|\mathbf{q}^\top \mu - 1\right| \geq t\right) \leq 2 \exp\left(-\frac{t^2}{2\mu^\top \Sigma \mu}\right). \tag{8}$$

*3. Non-Negativity of the Difference in Similarities:*

$$\mathbf{q}\top\mu - \mathbf{q}\top\mathbf{d} = \mathbf{q}^\top \mu(1 - \cos(\theta)) > 0. \tag{9}$$

*Remark* 1. The theorem provides a robust theoretical foundation for the QAEncoder's capabilities in computing the similarity between documents and queries. These concentration inequalities in Equation 7 and Equation 8 show that cosine similarities between $\mathbf{q}$, $\mathbf{d}$, and $\mu$ are concentrated around their expected values. Inequality 9 indicates that the similarity between the $\mathbf{q}$ and $\mu$ is always greater than between $\mathbf{q}$ and $\mathbf{d}$. This confirms that using the mean vector as a projection in QAEncoder better captures the semantic relationship between queries and documents. This theoretical result aligns with the experimental findings in Fig. 1, further validating QAEncoder's effectiveness.

### A.7.2 PROOF OF SIMILARITY BOUNDS

*Proof.* Given the setup where $\mathbf{q} \sim \mathcal{N}(\mu, \Sigma)$ is an $r$-dimensional Gaussian random vector with mean $\mu$ and covariance $\Sigma$, and the angle between another unit vector $\mathbf{d}$ and $\mu$ is $\theta$.

The cosine of the angle between $\mathbf{q}$ and $\mathbf{d}$ is given by:

$$\cos(\phi_{qd}) = \mathbf{q}^\top \mathbf{d}.$$

Since $\mathbf{q}$ is Gaussian, based on Lemma 1, the inner product $\mathbf{q}^\top \mathbf{d}$ is a linear transformation of $\mathbf{q}$ and hence is a normal distribution with mean $\mu^\top \mathbf{d}$ and variance $\mathbf{d}^\top \Sigma \mathbf{d}$. Given that $\mu$ and $\mathbf{d}$ are unit vectors and the angle between $\mu$ and $\mathbf{d}$ is $\theta$, we have:

$$\mu^\top \mathbf{d} = \cos(\theta).$$

Thus, $\mathbf{q}^\top \mathbf{d}$ can be approximated as:

$$\mathbf{q}^\top \mathbf{d} \sim \mathcal{N}(\cos(\theta), \mathbf{d}^\top \Sigma \mathbf{d}).$$

The concentration inequality for the cosine value between $\mathbf{q}$ and $\mathbf{d}$ follows from Hoeffding's inequality for zero-mean sub-Gaussian random variables, which can be expressed as:

$$\mathbb{P}\left(\left|\mathbf{q}^\top \mathbf{d} - \cos(\theta)\right| \geq t\right) \leq 2 \exp\left(-\frac{t^2}{2\mathbf{d}^\top \Sigma \mathbf{d}}\right).$$

Similarly, the cosine of the angle between between $\mathbf{q}$ and $\mu$ can be expressed as:

$$\cos(\phi_{q\mu}) = \mathbf{q}^\top \mu.$$

Given that $\mathbf{q} \sim \mathcal{N}(\mu, \Sigma)$, based on Lemma 1, the inner product $\mathbf{q}^\top \mu$, representing the cosine of the angle between $\mathbf{q}$ and $\mu$ is a linear transformation of a Gaussian random vector with mean:

$$\mathbb{E}[\mathbf{q}^\top \mu] = \mu^\top \mu = 1,$$

since $\mu$ is a unit vector, and variance:

$$\mathrm{Var}[\mathbf{q}^\top \mu] = \mu^\top \Sigma \mu.$$

Thus, $\mathbf{q}^\top \mu$ can be approximated as:

$$\mathbf{q}^\top \mu \sim \mathcal{N}(1, \mu^\top \Sigma \mu).$$

Applying the similar Hoeffding's inequality, the concentration inequality can be derived similarly:

$$\mathbb{P}\left(\left|\mathbf{q}^\top \mu - 1\right| \geq t\right) \leq 2 \exp\left(-\frac{t^2}{2\mu^\top \Sigma \mu}\right).$$

Notably, we observe that:

$$\mathbf{q}^\top \mathbf{d} = (\mathbf{q}^\top \mu)(\mu^\top \mathbf{d}) = (\mathbf{q}^\top \mu)\cos(\theta).$$

Therefore, comparing $\mathbf{q}^\top \mu$ and $\mathbf{q}^\top \mathbf{d}$, we find:

$$\mathbf{q}^\top \mu - \mathbf{q}^\top \mathbf{d} = \mathbf{q}^\top \mu(1 - \cos(\theta)).$$

Since $\mathbf{q} \neq \mathbf{d}$, it follows that $\cos(\theta) < 1$. Moreover, $\mathbf{q}^\top \mu > 0$ because $\mathbf{q}$ is a Gaussian random vector centered at $\mu$, which implies that $\mathbf{q}$ generally aligns positively with its mean $\mu$. Given that both $1 - \cos(\theta) > 0$ and $\mathbf{q}^\top \mu > 0$, we have:

$$\mathbf{q}^\top \mu - \mathbf{q}^\top \mathbf{d} = \mathbf{q}^\top \mu(1 - \cos(\theta)) > 0.$$

$\square$

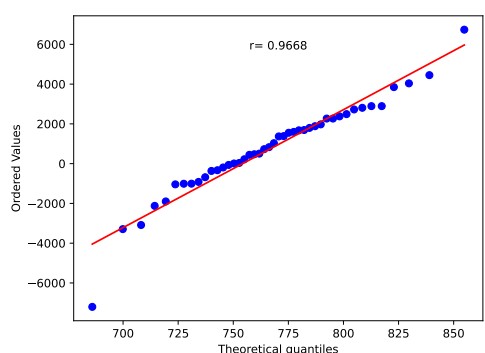

Figure 5: Q-Q plot against chi-squared distribution.

### A.7.3 NORMALITY TEST

To assess whether potential queries follow a Gaussian distribution in the embedding space, we employ two statistical tests: the Chi-Squared Q-Q Plot and the Anderson-Darling Test.

**Chi-Squared Q-Q Plot Verification**  We employ the Chi-Squared Q-Q Plot to assess whether the squared Mahalanobis distances 1 conform to the chi-squared distribution. By leveraging Lemma 2, We compare the observed $D^2$ values with the theoretical quantiles of the chi-squared distribution to assess the conformity of the data to a high-dimensional Gaussian model. We conclude that close alignment of the sample points along the 45-degree reference line indicates support for the original hypothesis.

Direct applications of high-dimensional normality tests, such as the Henze-Zirkler test, often lead to Type I errors in our case. E.g., testing on 768-dimensional normal samples revealed that Henze-Zirkler test demands high ratios of sample size to dimensionality, and it particularly susceptible to Type I errors when the sample size is not sufficiently large. Hence, we opted to perform uni-variate normality assessments on marginal distribution instead.

**Anderson-Darling Test**  To further evaluate the normality of marginal distributions, we conduct the Anderson-Darling test across all dimensions of the embeddings. The specific steps are as follows:

1. Null Hypothesis:
   - $H_0$: Each dimension's marginal distribution follows a Gaussian distribution.
   - $H_1$: At least one dimension's marginal distribution does not follow a Gaussian distribution.

2. Testing Results:

   Following the Anderson-Darling test, the results across all dimensions failed to reject $H_0$. This suggests that each dimension's marginal distribution can statistically be considered Gaussian, thereby supporting our hypothesis that potential queries conform to a high-dimensional Gaussian distribution.

In summary, through the verification of squared Mahalanobis distances using the Chi-Squared Q-Q Plot and the evaluation of marginal distributions with the Anderson-Darling test, we validate the plausibility that potential queries conform to a high-dimensional Gaussian distribution within the embedding space.

### A.7.4 SOME PROPERTIES OF GAUSSIAN DISTRIBUTION

The statement and proof of our main results contain some mathematical concepts. This section introduces these concepts, covering the fundamental lemmas and definitions essential for understanding our analysis.

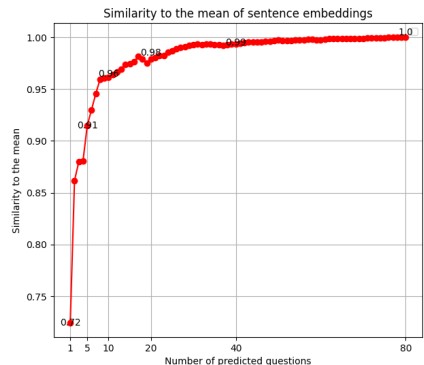

Figure 6: The convergence speed for Monte Carlo estimation of $\text{QAE}_{\text{base}}$, $n$ denotes the number of prediction queries. For documents with a length greater than 150 words from MSMARCO datasets, generating 10 queries exhibits a similarity score of 0.96 compared to generating 80 queries. Besides, the document fingerprint strategies introduces the hyperparameter $\alpha$ to $\text{QAE}_{\text{base}}$, further reducing the variance.

**Lemma 1.** *(Tong, 2012) Let* $\mathbf{x}$ *follow a multivariate Gaussian distribution:*

$$\mathbf{x} \sim \mathcal{N}(\mu, \Sigma),$$

*where $\mu$ is the mean vector and $\Sigma$ is the covariance matrix. For any linear transformation $A\mathbf{x} + \mathbf{b}$, the result is also multivariate normal:*

$$\mathbf{y} = A\mathbf{x} + \mathbf{b} \sim \mathcal{N}(A\mu + \mathbf{b}, A\Sigma A^{\top}).$$

*Remark* 2. The lemma shows that multivariate Gaussians remain Gaussian under linear transformations.

*Definition* 1. (Squared Mahalanobis Distance (McLachlan, 1999)) Assuming $\mathbf{x} \sim \mathcal{N}(\mu, \Sigma)$ in a high-dimensional Gaussian distribution, the squared Mahalanobis distance $D^2$ is defined as:

$$D^2 = (\mathbf{x} - \mu)^T \Sigma^{-1} (\mathbf{x} - \mu).$$

*Remark* 3. This metric quantifies the distance between the observed value and the mean.

*Lemma* 2. *(Chi-Squared Distribution of Squared Mahalanobis Distance (Mardia et al., 2024) ) If* $\mathbf{x}$ *is a $r$-dimensional Gaussian random variable, then the squared Mahalanobis distance $D^2$ follows a chi-squared distribution:*

$$D^2 \sim \chi_r^2,$$

*where $r$ denotes the dimensionality of the variable.*

*Remark* 4. The property that the squared Mahalanobis distance follows a chi-squared distribution can be viewed as a form of dimensionality reduction. By mapping a high-dimensional Gaussian variable to a scalar that encodes its deviation from the mean, adjusted for the covariance structure, this transformation reduces the complexity of the multivariate data while preserving key statistical properties in a single distance metric.

### A.8 MORE FIGURES AND TABLES

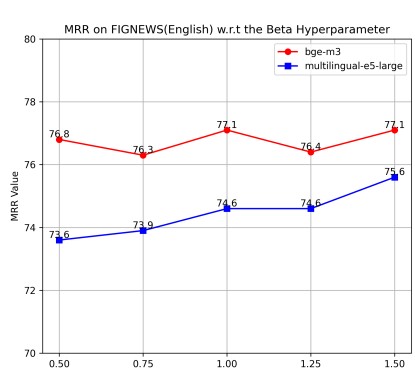

Figure 7: Ablation on $\beta$ hyperparameter for QAE$_{\text{txt}}$ on FIGNEWS(English) dataset.

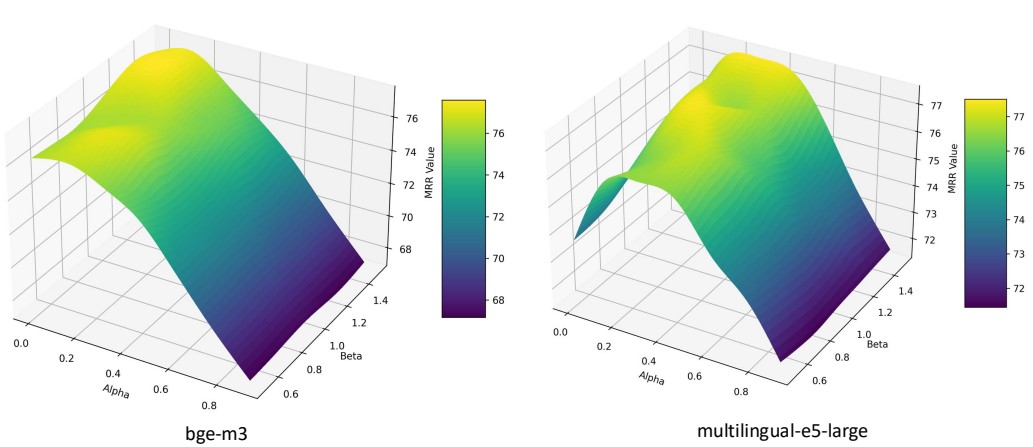

Figure 8: Ablation on $\alpha$ and $\beta$ hyperparameters for QAE$_{\text{hyb}}$ on FIGNEWS(English) dataset.

Table 5: Complete retrieval performance across six classical datasets: NQ, SQuAD, ELI5, Hot-PotQA, MSMARCO, and TriviaQA. Higher is better, with the best one is bolded. Hyperparameters including QAEncoder variants and weight terms $\alpha$, $\beta$ are optimized simultaneously for six classical datasets. '-' denotes default or null values.

| Model | Param | NQ | | | SQuAD | | | ELI5 | | | HotPotQA | | | MSMARCO | | | TriviaQA | | |
|---|---|---|---|---|---|---|---|---|---|---|---|---|---|---|---|---|---|---|---|
| | | MRR@8 | MAP@8 | NDCG@8 | MRR@8 | MAP@8 | NDCG@8 | MRR@8 | MAP@8 | NDCG@8 | MRR@8 | MAP@8 | NDCG@8 | MRR@8 | MAP@8 | NDCG@8 | MRR@8 | MAP@8 | NDCG@8 |
| **Sparse** | | | | | | | | | | | | | | | | | | | |
| BM25 | - | 17.9 | 10.4 | 12.8 | 46.2 | 46.2 | 48.2 | 12.6 | 9.6 | 11.5 | 13.8 | 9.9 | 11.1 | 53.2 | 53.1 | 62.1 | 9.8 | 6.7 | 7.8 |
| Doc2Query | - | 18.2 | 10.9 | 13.6 | 45.5 | 45.5 | 48.2 | 12.6 | 9.7 | 11.8 | 13.3 | 9.6 | 10.9 | 54.4 | 54.4 | 64.0 | 10.1 | 7.0 | 8.3 |
| DeepImpact | - | 21.8 | 12.9 | 15.7 | 47.8 | 47.9 | 50.0 | 12.9 | 9.9 | 11.9 | 14.4 | 10.4 | 11.6 | 64.4 | 64.1 | 71.8 | 11.8 | 8.2 | 9.5 |
| **Dense** | | | | | | | | | | | | | | | | | | | |
| bge-large-en-v1.5 | - | 87.0 | 83.2 | 68.0 | 76.2 | 76.2 | 79.9 | 57.7 | 55.8 | 55.8 | 93.4 | 90.1 | 83.3 | 75.4 | 75.4 | 81.3 | 78.4 | 75.0 | 67.5 |
| | $QAE_{hyb}$, $\alpha=0.15$, $\beta=0.5$ | 88.2 | 84.2 | 69.1 | 78.2 | 78.2 | 81.8 | 59.1 | 56.7 | 57.3 | 94.2 | 91.0 | 83.5 | 73.8 | 73.8 | 79.9 | 80.2 | 77.5 | 70.3 |
| multilingual-e5-large | - | 86.0 | 82.3 | 66.3 | 86.2 | 86.2 | 88.9 | 39.0 | 37.8 | 37.6 | 95.3 | 92.8 | 85.4 | 73.6 | 73.6 | 80 | 76.8 | 73.9 | 66.1 |
| | $QAE_{hyb}$, $\alpha=0.15$, $\beta=1.5$ | 86.1 | 81.9 | 66.7 | 84.9 | 84.9 | 87.9 | 46.4 | 45.6 | 43.3 | 94.3 | 91.0 | 83.4 | 70.1 | 70.1 | 77.3 | 79.3 | 77.0 | 68.9 |
| gte-base-en-v1.5 | - | 86.2 | 82.5 | 67.6 | 68.1 | 68.1 | 72.4 | 54.5 | 52.5 | 51.6 | 93.4 | 89.7 | 78.4 | 76.9 | 76.9 | 82.4 | 77.7 | 74.8 | 67.2 |
| | $QAE_{hyb}$, $\alpha=0.3$, $\beta=0.5$ | 85.5 | 81.6 | 67.4 | 74.8 | 74.8 | 78.7 | 57.0 | 54.8 | 55.1 | 92.9 | 88.9 | 76.9 | 76.0 | 76.0 | 81.4 | 78.8 | 76.1 | 69.2 |
| jina-embeddings-v2-small-en | - | 82.4 | 79.3 | 63.3 | 69.5 | 69.5 | 73.6 | 54.3 | 52.4 | 51.6 | 92.9 | 88.3 | 78.1 | 65.3 | 65.3 | 73.3 | 74.6 | 72.7 | 64.4 |
| | $QAE_{hyb}$, $\alpha=0.15$, $\beta=0.5$ | 83.3 | 79.9 | 64.1 | 72.4 | 72.4 | 76.2 | 53.8 | 52.3 | 51.6 | 91.7 | 87.3 | 76.6 | 68.2 | 68.2 | 75.7 | 75.9 | 73.9 | 65.5 |
| contriever | - | 78.8 | 75.0 | 60.8 | 64.8 | 64.8 | 69.4 | 51.3 | 49.2 | 49.8 | 89.0 | 84.6 | 74.1 | 55.4 | 55.4 | 63.6 | 71.2 | 68.6 | 61.4 |
| | $QAE_{emb}$, $\alpha=0.45$ | 84.0 | 80.2 | 65.7 | 74.9 | 74.9 | 78.9 | 55.7 | 54.0 | 54.4 | 89.8 | 86.2 | 75.1 | 67.1 | 67.1 | 74.2 | 76.4 | 73.9 | 67.4 |
| mcontriever | - | 52.1 | 48.7 | 37.6 | 49.3 | 49.3 | 54.9 | 43.0 | 41.2 | 40.6 | 83.0 | 78.3 | 67.9 | 49.6 | 49.6 | 56.7 | 59.4 | 56.8 | 49.7 |
| | $QAE_{emb}$, $\alpha=0.45$, $\beta=0.75$ | 61.4 | 58.5 | 45.9 | 64.7 | 64.7 | 69.5 | 51.0 | 49.1 | 48.3 | 85.2 | 80.9 | 69.5 | 65.1 | 65.1 | 71.8 | 70.4 | 67.6 | 60.4 |
| bce-embedding-base-v1 | - | 74.4 | 70.0 | 55.5 | 77.2 | 77.2 | 81.3 | 47.3 | 46.1 | 45.0 | 83.0 | 79.2 | 67.8 | 70.9 | 70.9 | 77.7 | 67.8 | 65.2 | 57.3 |
| | $QAE_{emb}$, $\alpha=0.3$ | 76.4 | 72.3 | 56.8 | 77.1 | 77.1 | 81.1 | 50.1 | 48.4 | 48.4 | 84.1 | 79.7 | 68.0 | 71.2 | 71.2 | 78 | 69.4 | 66.5 | 58.4 |
| text2vec-base-multilingual | - | 53.8 | 51.1 | 36.6 | 40.9 | 40.9 | 45.6 | 38.2 | 37.0 | 34.7 | 51.2 | 49.0 | 37.8 | 52.4 | 52.4 | 59.8 | 41.8 | 40.1 | 33.3 |
| | $QAE_{emb}$, $\alpha=0.6$, $\beta=0.5$ | 66.6 | 63.7 | 47.2 | 56.2 | 56.2 | 60.6 | 43.4 | 42.1 | 40.8 | 72.0 | 69.2 | 52.7 | 67.8 | 67.8 | 74.2 | 57.2 | 55.5 | 47.1 |
| dpr-multiset-base | - | 77.3 | 72.4 | 60.3 | 59.3 | 59.3 | 64.9 | 59.6 | 57.6 | 57.3 | 82.6 | 77.2 | 69.7 | 55.8 | 55.8 | 65 | 70.7 | 68.1 | 60.5 |
| | $QAE_{emb}$, $\alpha=0.45$ | 82.2 | 77.3 | 64.2 | 64.2 | 64.2 | 69.1 | 60.1 | 58 | 58.3 | 86.9 | 81.9 | 72.1 | 67.6 | 67.6 | 75.1 | 74.4 | 71.7 | 64.8 |
| dpr-single-nq-base | - | 77.6 | 72.8 | 62.0 | 60.3 | 60.3 | 66.1 | 57.3 | 55.2 | 55.4 | 75.3 | 70.8 | 62.9 | 61.5 | 61.5 | 70.1 | 65.6 | 62 | 56.1 |
| | $QAE_{emb}$, $\alpha=0.45$ | 81.8 | 76.7 | 63.3 | 66.4 | 66.4 | 70.8 | 58.8 | 56.5 | 57.2 | 81.6 | 77 | 67.5 | 68.4 | 68.4 | 75.5 | 70.3 | 66.7 | 60.2 |
| **Ablation** | | | | | | | | | | | | | | | | | | | |
| bge-large-en-v1.5 | - | 87.0 | 83.2 | 68.0 | 76.2 | 76.2 | 79.9 | 57.7 | 55.8 | 55.8 | 93.4 | 90.1 | 83.3 | 75.4 | 75.4 | 81.3 | 78.4 | 75.0 | 67.5 |
| | $\alpha=0.3$ | 87.4 | 83.3 | 68.2 | 78.0 | 78.0 | 81.6 | 59.2 | 56.9 | 57.0 | 93.6 | 90.3 | 83.1 | 74.7 | 74.7 | 80.5 | 79.4 | 76.4 | 69.4 |
| | $\beta=0.5$ | 88.1 | 84.3 | 68.9 | 77.6 | 77.6 | 81.5 | 58.1 | 55.9 | 56.6 | 94.2 | 91.0 | 83.6 | 73 | 73 | 79.5 | 80.1 | 77.2 | 70.0 |
| | $\alpha=0.15$, $\beta=0.5$ | 88.2 | 84.2 | 69.1 | 78.2 | 78.2 | 81.8 | 59.1 | 56.7 | 57.3 | 94.2 | 91.0 | 83.5 | 73.8 | 73.8 | 79.9 | 80.2 | 77.5 | 70.3 |
| multilingual-e5-large | - | 86.0 | 82.3 | 66.3 | 86.2 | 86.2 | 88.9 | 37.8 | 37.6 | 37.6 | 95.3 | 92.8 | 85.4 | 73.6 | 73.6 | 80 | 76.8 | 73.9 | 66.1 |
| | $\alpha=0.3$ | 87.0 | 82.9 | 66.9 | 85.3 | 85.3 | 88.0 | 44.8 | 43.5 | 42.6 | 94.8 | 92.2 | 84.5 | 69.9 | 69.9 | 77.2 | 78.5 | 75.7 | 67.6 |
| | $\beta=1.5$ | 86.1 | 82.2 | 66.5 | 84.5 | 84.5 | 87.5 | 45.2 | 44.1 | 42.2 | 94.9 | 91.5 | 84.0 | 69.7 | 69.7 | 77 | 79.6 | 77.3 | 69.0 |
| | $\alpha=0.15$, $\beta=1.5$ | 86.1 | 81.9 | 66.7 | 84.9 | 84.9 | 87.9 | 46.4 | 45.6 | 43.3 | 94.3 | 91.0 | 83.4 | 70.1 | 70.1 | 77.3 | 79.3 | 77.0 | 68.9 |
| gte-base-en-v1.5 | - | 86.2 | 82.5 | 67.6 | 68.1 | 68.1 | 72.4 | 54.5 | 52.5 | 51.6 | 93.4 | 89.7 | 78.4 | 76.9 | 76.9 | 82.4 | 77.7 | 74.8 | 67.2 |
| | $\alpha=0.3$ | 85.9 | 82.2 | 67.9 | 72.4 | 72.4 | 76.3 | 56.7 | 54.3 | 54.3 | 92.6 | 89.0 | 77.0 | 77.2 | 77.2 | 82.3 | 78.6 | 76.0 | 68.8 |
| | $\beta=0.5$ | 85.3 | 81.3 | 67.0 | 73.0 | 73.0 | 77.2 | 55.6 | 53.4 | 53.4 | 93.3 | 90.0 | 77.7 | 76.7 | 76.7 | 82.3 | 78.6 | 75.8 | 68.7 |
| | $\alpha=0.3$, $\beta=0.5$ | 85.5 | 81.6 | 67.4 | 74.8 | 74.8 | 78.7 | 57.0 | 54.8 | 55.1 | 92.9 | 88.9 | 76.9 | 76.0 | 76.0 | 81.4 | 78.8 | 76.1 | 69.2 |
| jina-embeddings-v2-small-en | - | 82.4 | 79.3 | 63.3 | 69.5 | 69.5 | 73.6 | 54.3 | 52.4 | 51.6 | 92.9 | 88.3 | 78.1 | 65.3 | 65.3 | 73.3 | 74.6 | 72.7 | 64.4 |
| | $\alpha=0.3$ | 83.0 | 79.5 | 63.5 | 71.2 | 71.2 | 75.2 | 54.6 | 52.7 | 52.1 | 91.5 | 87.1 | 76.4 | 65.3 | 65.3 | 73.5 | 76.2 | 74.0 | 65.6 |
| | $\beta=0.5$ | 82.8 | 79.6 | 63.8 | 71.5 | 71.5 | 75.3 | 53.5 | 51.9 | 51.1 | 92.1 | 87.6 | 77.3 | 69.4 | 69.4 | 76.6 | 75.6 | 73.4 | 65.2 |
| | $\alpha=0.15$, $\beta=0.5$ | 83.3 | 79.9 | 64.1 | 72.4 | 72.4 | 76.2 | 53.8 | 52.3 | 51.6 | 91.7 | 87.3 | 76.6 | 68.2 | 68.2 | 75.7 | 75.9 | 73.9 | 65.5 |
| contriever | - | 78.8 | 75.0 | 60.8 | 64.8 | 64.8 | 69.4 | 51.3 | 49.2 | 49.8 | 89.0 | 84.6 | 74.1 | 55.4 | 55.4 | 63.6 | 71.2 | 68.6 | 61.4 |
| | $\alpha=0.45$ | 84.0 | 80.2 | 65.7 | 74.9 | 74.9 | 78.9 | 55.7 | 54.0 | 54.4 | 89.8 | 86.2 | 75.1 | 67.1 | 67.1 | 74.2 | 76.4 | 73.9 | 67.4 |
| | $\beta=1.25$ | 79.7 | 76.5 | 61.0 | 64.4 | 64.4 | 69.1 | 50.0 | 48.4 | 47.7 | 91.5 | 87.0 | 76.6 | 64 | 64 | 71.7 | 70.2 | 67.6 | 59.3 |
| | $\alpha=0.45$, $\beta=0.5$ | 84.0 | 80.3 | 65.5 | 72.5 | 72.5 | 77.0 | 55.1 | 53.3 | 53.8 | 89.7 | 85.2 | 75.1 | 67 | 67 | 73.6 | 75.4 | 72.6 | 65.6 |
| mcontriever | - | 52.1 | 48.7 | 37.6 | 49.3 | 49.3 | 54.9 | 43.0 | 41.2 | 40.6 | 83.0 | 78.3 | 67.9 | 49.6 | 49.6 | 56.7 | 59.4 | 56.8 | 49.7 |
| | $\alpha=0.45$ | 62.1 | 59.3 | 46.5 | 63.6 | 63.6 | 68.6 | 49.2 | 47.5 | 47.2 | 85.7 | 81.5 | 70.4 | 64.5 | 64.5 | 71.8 | 70.1 | 67.1 | 60.3 |
| | $\beta=0.5$ | 58.3 | 51.0 | 38.3 | 53.8 | 53.8 | 58.7 | 42.1 | 40.2 | 39.4 | 86.2 | 81.9 | 70.1 | 58.3 | 58.3 | 66 | 62.0 | 59.6 | 51.9 |
| | $\alpha=0.45$, $\beta=0.5$ | 61.4 | 58.5 | 45.9 | 64.7 | 64.7 | 69.5 | 51.0 | 49.1 | 48.3 | 85.2 | 80.9 | 69.5 | 65.1 | 65.1 | 71.8 | 70.4 | 67.6 | 60.4 |
| bce-embedding-base-v1 | - | 74.4 | 70.0 | 55.5 | 77.2 | 77.2 | 81.3 | 47.3 | 46.1 | 45.0 | 83.0 | 79.2 | 67.8 | 70.9 | 70.9 | 77.7 | 67.8 | 65.2 | 57.3 |
| | $\alpha=0.45$ | 76.4 | 72.3 | 56.8 | 77.1 | 77.1 | 81.1 | 50.1 | 48.4 | 48.4 | 84.1 | 79.7 | 68.0 | 71.2 | 71.2 | 78 | 69.4 | 66.5 | 58.4 |
| | $\beta=0.5$ | 75.0 | 70.5 | 55.7 | 72.6 | 72.6 | 77.2 | 49.8 | 47.6 | 47.7 | 82.1 | 77.5 | 65.9 | 72.8 | 72.8 | 79.2 | 68.7 | 65.7 | 58.0 |
| | $\alpha=0.3$, $\beta=0.5$ | 76.9 | 72.8 | 57.2 | 73.7 | 73.7 | 77.9 | 51.4 | 49.3 | 49.4 | 82.3 | 77.5 | 65.7 | 69.9 | 69.9 | 76.9 | 69.7 | 66.4 | 58.7 |
| text2vec-base-multilingual | - | 53.8 | 51.1 | 36.6 | 40.9 | 40.9 | 45.6 | 38.2 | 37.0 | 34.7 | 51.2 | 49.0 | 37.8 | 52.4 | 52.4 | 59.8 | 41.8 | 40.1 | 33.3 |
| | $\alpha=0.75$ | 66.7 | 63.5 | 47.4 | 54.8 | 54.8 | 59.3 | 42.9 | 41.7 | 40.7 | 71.8 | 69.1 | 52.8 | 69.1 | 69.1 | 75.4 | 57.8 | 56.1 | 47.6 |
| | $\beta=0.5$ | 56.6 | 54.1 | 40.1 | 46.5 | 46.5 | 51.3 | 39.0 | 37.9 | 36.2 | 61.3 | 59.2 | 45.2 | 57.2 | 57.2 | 64.7 | 47.0 | 45.5 | 38.2 |
| | $\alpha=0.6$, $\beta=0.5$ | 66.6 | 63.7 | 47.2 | 56.2 | 56.2 | 60.6 | 43.4 | 42.1 | 40.8 | 72.0 | 69.2 | 52.7 | 67.8 | 67.8 | 74.2 | 57.2 | 55.5 | 47.1 |

Table 6: Comprehensive retrieval performance on the latest datasets FIGNEWS and CRUD-RAG. Higher is better, with the best one bolded. Hyperparameters including QAEncoder variants and weight terms $\alpha$, $\beta$ are optimized simultaneously for six latest datasets. '-' denotes default or null values.

| Model | Param | FIGNEWS(English) | | FIGNEWS(Arabic) | | CRUD-RAG(Chinese) | | FIGNEWS(French) | | FIGNEWS(Hindi) | | FIGNEWS(Hebrew) | |
|---|---|---|---|---|---|---|---|---|---|---|---|---|---|
| | | MRR@8 | NDCG@8 | MRR@8 | NDCG@8 | MRR@8 | NDCG@8 | MRR@8 | NDCG@8 | MRR@8 | NDCG@8 | MRR@8 | NDCG@8 |
| bge-m3 | - | 74.3 | 78.2 | 77.7 | 80.5 | 86.8 | 88.9 | 73.4 | 76.9 | 58.4 | 63.8 | 78.6 | 81 |
| | QAE$_{txt}$, $\beta = 1.5$ | **77.1** | **80.8** | **80** | **82.8** | **89** | **90.8** | **76.7** | **79.9** | **62.5** | **67.5** | **79.8** | **82.4** |
| multilingual-e5-small | - | 70.9 | 74.7 | 74.0 | 77.1 | 81.7 | 84.2 | 72.8 | 76.1 | 66.6 | 70.1 | 52.3 | 57.2 |
| | QAE$_{hyb}$, $\alpha = 0.3$, $\beta = 0.5$ | **74.4** | **78.2** | **78.8** | **81.2** | **87.4** | **89.2** | **74.1** | **77.6** | **59.4** | **64.0** | **77.4** | **80.1** |
| multilingual-e5-base | - | 74.7 | 77.9 | 72.2 | 75.7 | 86.1 | 88.1 | 70.9 | 74.3 | 57.6 | 62.4 | 72.6 | 75.6 |
| | QAE$_{emb}$, $\alpha = 0.3$ | **77.4** | **80.7** | **77.1** | **80** | **88.5** | **90.5** | **76.6** | **79.8** | **61.3** | **66** | **77.7** | **80.4** |
| multilingual-e5-large | - | 73.8 | 77.5 | 76.6 | 79.9 | 85.7 | 88.1 | 70.4 | 74 | 52.7 | 58.4 | 73.9 | 77.1 |
| | QAE$_{hyb}$, $\alpha = 0.15$, $\beta = 1.25$ | **77** | **80.4** | **82.1** | **84.9** | **89.1** | **91.1** | **77.2** | **80.4** | **60.3** | **65** | **77.6** | **80.6** |
| gte-multilingual-base | - | 65.3 | 69.9 | 73.2 | 76.2 | 82.9 | 85.6 | 62.8 | 67 | 52.1 | 57.8 | 65.9 | 69.3 |
| | QAE$_{hyb}$, $\alpha = 0.15$, $\beta = 1.5$ | **75.3** | **79** | **76.2** | **79** | **85.5** | **88** | **66.6** | **70.7** | **55.8** | **61.4** | **73.9** | **77.2** |
| mcontriever | - | 32.8 | 36.3 | 40.1 | 44.2 | 71.7 | 75.8 | 34.7 | 38.6 | 26.8 | 31.4 | 49.2 | 53.7 |
| | QAE$_{hyb}$, $\alpha = 0.45$, $\beta = 1.25$ | **61.2** | **65.4** | **68.2** | **71.9** | **88.7** | **90.5** | **64.5** | **68.5** | **50.3** | **55.6** | **69.8** | **73** |
| bce-embedding-base-v1 | - | 58.9 | 63.3 | - | - | 76.8 | 80.3 | - | - | - | - | - | - |
| | QAE$_{hyb}$, $\alpha = 0.3$, $\beta = 0.5$ | **66.6** | **70.6** | - | - | **85.9** | **88.1** | - | - | - | - | - | - |
| text2vec-base-multilingual | - | 38.5 | 43 | 27.7 | 31.4 | 12.1 | 13.7 | 33.2 | 37.5 | 15.3 | 18.9 | 12.3 | 14.6 |
| | QAE$_{emb}$, $\alpha = 0.75$ | **55.2** | **59.3** | **51.3** | **54.8** | **55.3** | **58.6** | **49.1** | **53.8** | **35.9** | **40.6** | **46.7** | **51** |
| **Ablation** | | | | | | | | | | | | | |
| bge-m3 | - | 74.3 | 78.2 | 77.7 | 80.5 | 86.8 | 88.9 | 73.4 | 76.9 | 58.4 | 63.8 | 78.6 | 81.0 |
| | $\alpha = 0.3$ | 76.2 | 80.0 | 80.0 | 82.6 | 88.7 | 90.6 | 75.0 | 78.5 | 60.9 | 65.9 | 78.9 | 81.7 |
| | $\beta = 1.5$ | 77.1 | 80.8 | 80.1 | 82.8 | 89.0 | 90.8 | 76.7 | 79.9 | 62.5 | 67.5 | 79.8 | 82.4 |
| | $\alpha = 0.15$, $\beta = 1.5$ | 77.3 | 80.7 | 80.6 | 83.3 | 89.4 | 91.2 | 76.4 | 79.7 | 61.5 | 66.3 | 79.7 | 82.4 |
| multilingual-e5-small | - | 70.9 | 74.7 | 74.0 | 77.1 | 81.7 | 84.2 | 72.8 | 76.1 | 66.6 | 70.1 | 52.3 | 57.2 |
| | $\alpha = 0.45$ | 74.6 | 78.3 | 76.9 | 79.6 | 87.8 | 89.7 | 73.5 | 76.9 | 57.9 | 62.7 | 76.9 | 79.6 |
| | $\beta = 1.0$ | 73.1 | 76.9 | 79.1 | 81.6 | 85.0 | 87.1 | 70.4 | 74.3 | 58.5 | 63.3 | 77.0 | 79.9 |
| | $\alpha = 0.3$, $\beta = 0.5$ | 74.4 | 78.2 | 78.8 | 81.2 | 87.4 | 89.2 | 74.1 | 77.6 | 59.4 | 64.0 | 77.4 | 80.1 |
| multilingual-e5-base | - | 74.7 | 77.9 | 72.2 | 75.7 | 86.1 | 88.1 | 70.9 | 74.3 | 57.6 | 62.4 | 72.6 | 75.6 |
| | $\alpha = 0.3$ | 77.4 | 80.7 | 77.1 | 80.0 | 88.5 | 90.5 | 76.6 | 79.8 | 61.3 | 66.0 | 77.7 | 80.4 |
| | $\beta = 0.75$ | 74.5 | 78.3 | 76.2 | 79.2 | 87.5 | 89.6 | 72.3 | 76.2 | 62.3 | 67.1 | 77.0 | 79.7 |
| | $\alpha = 0.3$, $\beta = 0.5$ | 76.5 | 80.1 | 77.3 | 80.3 | 89.0 | 90.8 | 75.3 | 78.9 | 62.5 | 67.1 | 77.1 | 79.8 |
| multilingual-e5-large | - | 73.8 | 77.5 | 76.6 | 79.9 | 85.7 | 88.1 | 70.4 | 74.0 | 52.7 | 58.4 | 73.9 | 77.1 |
| | $\alpha = 0.45$ | 77.8 | 81.1 | 79.7 | 82.7 | 89.7 | 91.4 | 77.0 | 79.9 | 58.2 | 63.1 | 78.0 | 80.9 |
| | $\beta = 1.5$ | 75.6 | 79.0 | 80.7 | 83.7 | 88.3 | 90.4 | 76.0 | 79.3 | 59.7 | 64.8 | 77.0 | 79.7 |
| | $\alpha = 0.15$, $\beta = 1.25$ | 77.0 | 80.4 | 82.1 | 84.9 | 89.1 | 91.1 | 77.2 | 80.4 | 60.3 | 65.1 | 77.6 | 80.6 |
| gte-multilingual-base | - | 65.3 | 69.9 | 73.2 | 76.2 | 82.9 | 85.6 | 62.8 | 67.0 | 52.1 | 57.8 | 65.9 | 69.3 |
| | $\alpha = 0.45$ | 69.0 | 73.1 | 76.8 | 79.3 | 87.3 | 89.4 | 66.8 | 70.4 | 52.9 | 58.0 | 72.3 | 75.6 |
| | $\beta = 1.5$ | 75.5 | 79.1 | 75.7 | 78.5 | 84.4 | 87.1 | 65.9 | 70.3 | 56.5 | 61.8 | 72.7 | 76.1 |
| | $\alpha = 0.15$, $\beta = 1.5$ | 75.3 | 79.0 | 76.2 | 79.0 | 85.5 | 88.0 | 66.6 | 70.7 | 55.8 | 61.4 | 73.9 | 77.2 |
| mcontriever | - | 32.8 | 36.3 | 40.1 | 44.2 | 71.7 | 75.8 | 34.7 | 38.6 | 26.8 | 31.4 | 49.2 | 53.7 |
| | $\alpha = 0.6$ | 58.6 | 63.2 | 67.1 | 71.0 | 88.3 | 90.1 | 62.5 | 66.2 | 50.2 | 55.1 | 69.4 | 72.5 |
| | $\beta = 1.5$ | 48.7 | 52.8 | 59.3 | 63.3 | 79.9 | 83.1 | 53.7 | 57.7 | 44.9 | 49.6 | 60.6 | 64.2 |
| | $\alpha = 0.45$, $\beta = 1.25$ | 61.2 | 65.4 | 68.2 | 71.9 | 88.7 | 90.5 | 64.5 | 68.5 | 50.3 | 55.6 | 69.8 | 73.0 |
| bce-embedding-base-v1 | - | 58.9 | 63.3 | - | - | 76.8 | 80.3 | - | - | - | - | - | - |
| | $\alpha = 0.45$ | 66.6 | 70.7 | - | - | 85.2 | 87.5 | - | - | - | - | - | - |
| | $\beta = 1.5$ | 64.0 | 68.1 | - | - | 82.3 | 84.9 | - | - | - | - | - | - |
| | $\alpha = 0.3$, $\beta = 0.5$ | 66.6 | 70.6 | - | - | 85.9 | 88.1 | - | - | - | - | - | - |
| text2vec-base-multilingual | - | 38.5 | 43.0 | 27.7 | 31.4 | 12.1 | 13.7 | 33.2 | 37.5 | 15.3 | 18.9 | 12.3 | 14.6 |
| | $\alpha = 0.75$ | 55.2 | 59.3 | 51.3 | 54.8 | 55.3 | 58.6 | 49.1 | 53.8 | 35.9 | 40.6 | 46.7 | 51.0 |
| | $\beta = 1.5$ | 46.0 | 50.3 | 35.3 | 38.8 | 21.1 | 23.2 | 42.9 | 46.8 | 21.1 | 24.4 | 18.5 | 21.3 |
| | $\alpha = 0.75$, $\beta = 0.5$ | 55.0 | 59.2 | 50.2 | 53.7 | 57.7 | 60.4 | 48.9 | 53.1 | 34.2 | 38.9 | 47.0 | 50.9 |

Table 7: Complete performance comparison of QAEncoder variants on latest datasets FIGNEWS and CRUD-RAG. Higher is better, with the best one bolded. Hyperparameters are optimized simultaneously across the six latest datasets. $n$ indicates the number of predicted queries in QA$_{naive}$.

| Model | Param | FIGNEWS(English) | | FIGNEWS(Arabic) | | CRUD-RAG(Chinese) | | FIGNEWS(French) | | FIGNEWS(Hindi) | | FIGNEWS(Hebrew) | |
|---|---|---|---|---|---|---|---|---|---|---|---|---|---|
| | | MRR@8 | NDCG@8 | MRR@8 | NDCG@8 | MRR@8 | NDCG@8 | MRR@8 | NDCG@8 | MRR@8 | NDCG@8 | MRR@8 | NDCG@8 |
| bge-m3 | QAE$_{emb}$, $\alpha = 0.3$ | 76.2 | 80.0 | 80.0 | 82.6 | 88.7 | 90.6 | 75.0 | 78.5 | 60.9 | 65.9 | 78.9 | 81.7 |
| | QAE$_{txt}$, $\beta = 1.5$ | 77.1 | **80.8** | 80.1 | 82.8 | 89.0 | 90.8 | **76.7** | **79.9** | **62.5** | **67.5** | **79.8** | **82.4** |
| | QAE$_{hyb}$, $\alpha = 0.15$, $\beta = 1.5$ | **77.3** | 80.7 | **80.6** | **83.3** | **89.4** | **91.2** | 76.4 | 79.7 | 61.5 | 66.3 | 79.7 | 82.4 |
| | $n = 10$ | 76.8 | 79.5 | 76.9 | 79.2 | 85.9 | 87.8 | 71.7 | 73.8 | 62.2 | 65.8 | 68.0 | 71.3 |
| multilingual-e5-large | QAE$_{emb}$, $\alpha = 0.45$ | **77.8** | **81.1** | 79.7 | 82.7 | **89.7** | **91.4** | 77.0 | 79.9 | 58.2 | 63.1 | **78.0** | **80.9** |
| | QAE$_{txt}$, $\beta = 1.5$ | 75.6 | 79.0 | 80.7 | 83.7 | 88.3 | 90.4 | 76.0 | 79.3 | 59.7 | 64.8 | 77.0 | 79.7 |
| | QAE$_{hyb}$, $\alpha = 0.15$, $\beta = 1.25$ | 77.0 | 80.5 | **82.1** | **84.9** | 89.1 | 91.1 | **77.2** | **80.4** | 60.3 | 65.1 | 77.6 | 80.6 |
| | $n = 10$ | 76.8 | 79.4 | 76.5 | 79.2 | 84.9 | 86.8 | 70.4 | 73.1 | **61.5** | **65.4** | 69.3 | 71.8 |

Table 8: The table illustrates a comprehensive performance comparison of QAEncoder against training-based and document-centric methods on the latest datasets: FIGNEWS and CRUD-RAG. Higher is better, with the best one bolded. Hyperparameters $\alpha$, $\beta$ are optimized simultaneously across the six latest datasets. $n$ denotes the number of pseudo-documents in HyDE. '-' indicates default or null values.

| Model | Param | FIGNEWS(English) | | FIGNEWS(Arabic) | | CRUD-RAG(Chinese) | | FIGNEWS(French) | | FIGNEWS(Hindi) | | FIGNEWS(Hebrew) | |
|---|---|---|---|---|---|---|---|---|---|---|---|---|---|
| | | MRR@8 | NDCG@8 | MRR@8 | NDCG@8 | MRR@8 | NDCG@8 | MRR@8 | NDCG@8 | MRR@8 | NDCG@8 | MRR@8 | NDCG@8 |
| mcontriever-msmarco | - | 65.9 | 70.0 | 70.1 | 73.5 | 85.0 | 87.4 | 66.3 | 70.0 | 48.6 | 53.5 | 69.3 | 72.6 |
| | QAE$_{hyb}$, $\alpha = 0.3$, $\beta = 0.75$ | 72.1 | 76.3 | 77.2 | 80.2 | 88.6 | 90.6 | 74.2 | 77.7 | 58.7 | 63.5 | 76.3 | 79.4 |
| multilingual-e5-large-instruct | - | 66.7 | 70.7 | 74.9 | 78.1 | 79.8 | 82.5 | 65.8 | 70.1 | 48.0 | 53.1 | 69.5 | 72.8 |
| | QAE$_{hyb}$, $\alpha = 0.15$, $\beta = 1.5$ | 75.5 | 79.4 | 80.7 | 83.5 | 88.9 | 90.6 | 75.7 | 79.2 | 58.9 | 64.1 | 79.3 | 82.1 |
| mcontriever | - | 32.8 | 36.3 | 40.1 | 44.2 | 71.7 | 75.8 | 34.7 | 38.6 | 26.9 | 31.4 | 49.2 | 53.7 |
| | HyDE, $n = 8$ | 24.9 | 27.6 | 35.1 | 40.4 | 70.7 | 74.2 | 25.6 | 28.6 | 11.6 | 14.3 | 41.8 | 47.0 |
| multilingual-e5-large | - | 73.8 | 77.5 | 76.6 | 79.9 | 85.7 | 88.2 | 70.4 | 74.0 | 52.7 | 58.4 | 73.9 | 77.1 |
| | HyDE, $n = 8$ | 63.4 | 67.7 | 68.2 | 73.8 | 81.5 | 83.9 | 58.2 | 62.6 | 45.5 | 51.0 | 65.3 | 69.6 |

Table 9: Retrieval performance of **monolingual and bilingual embedding models** on the latest datasets FIGNEWS(English) and CRUD-RAG(Chinese). Higher is better, with the best one bolded. '-' denotes default or null values.

| Model | Param | FIGNEWS(English) | | FIGNEWS(Arabic) | | CRUD-RAG(Chinese) | |
|---|---|---|---|---|---|---|---|
| | | MRR@10 | NDCG@10 | MRR@10 | NDCG@10 | MRR@10 | NDCG@10 |
| e5-large-v2 | - | 73.2 | 77.2 | - | - | - | - |
| | QAE$_{hyb}$, $\alpha = 0.3$, $\beta = 1.0$ | **78.7** | **82** | - | - | - | - |
| jina-embeddings-v2-small-en | - | 65.4 | 69.9 | - | - | - | - |
| | QAE$_{hyb}$, $\alpha = 0.15$, $\beta = 1.25$ | **72.6** | **76.7** | - | - | - | - |
| jina-embeddings-v2-base-zh | - | 63.4 | 67.6 | - | - | 75 | 78.8 |
| | QAE$_{hyb}$, $\alpha = 0.45$, $\beta = 1.25$ | **66.7** | **71.1** | - | - | **86.3** | **88.4** |
| jina-embeddings-v2-base-en | - | 65.3 | 69.4 | - | - | - | - |
| | QAE$_{hyb}$, $\alpha = 0.3$, $\beta = 0.5$ | **72.4** | **76.5** | - | - | - | - |
| gte-base-en-v1.5 | - | 65.6 | 70.2 | - | - | - | - |
| | QAE$_{hyb}$, $\alpha = 0.15$, $\beta = 1.5$ | **71.3** | **75.6** | - | - | - | - |
| contriever | - | 49.6 | 54.5 | - | - | - | - |
| | QAE$_{hyb}$, $\alpha = 0.45$, $\beta = 1.25$ | **70.7** | **74.7** | - | - | - | - |
| bge-large-zh-v1.5 | - | - | - | - | - | 76.8 | 80 |
| | QAE$_{hyb}$, $\alpha = 0.45$, $\beta = 1.25$ | - | - | - | - | **89** | **90.7** |
| bge-large-zh | - | - | - | - | - | 74.3 | 77.8 |
| | QAE$_{hyb}$, $\alpha = 0.45$, $\beta = 1.5$ | - | - | - | - | **88.6** | **90.4** |
| bge-large-en-v1.5 | - | 66.4 | 71 | - | - | - | - |
| | QAE$_{hyb}$, $\alpha = 0.15$, $\beta = 1.5$ | **74.3** | **78.2** | - | - | - | - |
| bge-large-en | - | 61.9 | 66.4 | - | - | - | - |
| | QAE$_{hyb}$, $\alpha = 0.3$, $\beta = 0.75$ | **71.3** | **75.4** | - | - | - | - |
| bge-base-zh-v1.5 | - | - | - | - | - | 79.4 | 82.2 |
| | QAE$_{hyb}$, $\alpha = 0.6$, $\beta = 1.0$ | - | - | - | - | **88.9** | **90.6** |
| bge-base-zh | - | - | - | - | - | 73.7 | 76.7 |
| | QAE$_{hyb}$, $\alpha = 0.6$, $\beta = 1.5$ | - | - | - | - | **88.8** | **90.5** |
| bge-base-en-v1.5 | - | 66.4 | 70.5 | - | - | - | - |
| | QAE$_{txt}$, $\beta = 1.0$ | **74.1** | **77.5** | - | - | - | - |
| bge-base-en | - | 64.9 | 68.9 | - | - | - | - |
| | QAE$_{hyb}$, $\alpha = 0.3$, $\beta = 1.5$ | **71.7** | **75.9** | - | - | - | - |
| bce-embedding-base-v1 | - | 59.1 | 63.8 | - | - | 76.9 | 80.5 |
| | QAE$_{hyb}$, $\alpha = 0.3$, $\beta = 0.5$ | **66.8** | **71.1** | - | - | **86** | **88.3** |

Table 10: Complete retrieval performance across six classical datasets: NQ, SQuAD, ELI5, HotPotQA, MSMARCO, and TriviaQA (**Top-k = 10**). Higher is better, with the best one is bolded. Hyperparameters including QAEncoder variants and weight terms $\alpha$, $\beta$ are optimized simultaneously for six classical datasets. '-' denotes default or null values.

| Model | Param | NQ MRR@10 | MAP@10 | NDCG@10 | SQuAD MRR@10 | MAP@10 | NDCG@10 | ELI5 MRR@10 | MAP@10 | NDCG@10 | HotPotQA MRR@10 | MAP@10 | NDCG@10 | MSMARCO MRR@10 | MAP@10 | NDCG@10 | TriviaQA MRR@10 | MAP@10 | NDCG@10 |
|---|---|---|---|---|---|---|---|---|---|---|---|---|---|---|---|---|---|---|---|
| **Sparse** | | | | | | | | | | | | | | | | | | | |
| BM25 | - | 18.0 | 10.4 | 12.9 | 46.3 | 46.3 | 48.4 | 12.8 | 9.7 | 12.0 | 13.8 | 9.9 | 11.2 | 53.7 | 53.8 | 63.9 | 9.9 | 6.7 | 7.9 |
| Doc2Query | - | 18.2 | 11.0 | 13.9 | 45.6 | 45.6 | 48.5 | 12.8 | 9.9 | 12.2 | 13.3 | 9.6 | 11.0 | 54.9 | 54.9 | 65.5 | 10.1 | 7.0 | 8.4 |
| DeepImpact | - | 21.8 | 13.0 | 15.8 | 47.9 | 48.0 | 50.2 | 12.9 | 10.0 | 12.1 | 14.5 | 10.4 | 11.6 | 64.6 | 64.3 | 72.3 | 11.8 | 8.3 | 9.5 |
| **Dense** | | | | | | | | | | | | | | | | | | | |
| bge-large-en-v1.5 | - | 87.1 | 82.9 | 68.5 | 76.3 | 76.3 | 80.4 | 58.0 | 56.0 | 56.8 | 93.4 | 89.7 | 83.6 | **75.4** | **75.4** | **81.3** | 78.5 | 74.6 | 68.1 |
| | QAE$_{hyb}$, $\alpha=0.15$, $\beta=0.5$ | **88.2** | **83.8** | **69.5** | 78.4 | 78.4 | 82.5 | 59.4 | 56.8 | 58.2 | 94.2 | 90.7 | 83.6 | 73.9 | 73.9 | 80.2 | **80.3** | **77.2** | **70.8** |
| multilingual-e5-large | - | 86.0 | 81.4 | 66.8 | 86.4 | 86.4 | 89.3 | 39.4 | 38.0 | 38.7 | 95.3 | 92.2 | 85.7 | 73.7 | 73.7 | 80.3 | 76.9 | 73.6 | 66.6 |
| | QAE$_{hyb}$, $\alpha=0.15$, $\beta=1.5$ | 86.1 | 81.4 | 67.3 | 85.0 | 85.0 | 88.1 | 46.7 | 45.6 | 44.5 | 94.4 | 90.6 | 83.7 | 70.2 | 70.2 | 77.5 | 79.4 | 76.8 | 69.2 |
| gte-base-en-v1.5 | - | 86.2 | 81.7 | 68.2 | 68.4 | 68.4 | 73.2 | 54.7 | 52.5 | 52.3 | 93.4 | 89.1 | 78.6 | 77.0 | 77.0 | 82.7 | 77.7 | 74.2 | 67.6 |
| | QAE$_{hyb}$, $\alpha=0.3$, $\beta=0.5$ | 85.5 | 81.1 | 67.8 | 74.9 | 74.9 | 79.1 | 57.2 | 54.7 | 55.9 | 93.0 | 88.7 | 77.1 | 76.3 | 76.3 | 82.3 | 78.8 | 75.8 | 69.7 |
| jian-embeddings-v2-small-en | - | 82.4 | 78.7 | 63.7 | 69.7 | 69.7 | 74.3 | 54.5 | 52.2 | 52.5 | 92.9 | 87.6 | 78.5 | 65.5 | 65.5 | 73.9 | 74.7 | 72.5 | 64.7 |
| | QAE$_{hyb}$, $\alpha=0.15$, $\beta=0.5$ | 83.3 | 79.3 | 64.5 | 72.7 | 72.7 | 76.8 | 54.1 | 52.0 | 52.5 | 91.7 | 86.7 | 77.1 | 68.4 | 68.4 | 76.3 | 76.0 | 73.6 | 66.0 |
| contriever | - | 78.8 | 74.4 | 61.4 | 65.1 | 65.1 | 70.0 | 51.5 | 49.1 | 50.6 | 89.0 | 84.0 | 74.4 | 55.8 | 55.8 | 64.7 | 71.3 | 68.2 | 61.9 |
| | QAE$_{emb}$, $\alpha=0.45$ | 84.0 | 79.5 | 66.2 | 75.1 | 75.1 | 79.5 | 55.9 | 53.8 | 55.3 | 89.9 | 85.5 | 75.5 | 67.3 | 67.3 | 74.8 | 76.4 | 73.6 | 67.9 |
| mcontriever | - | 52.4 | 48.2 | 38.5 | 49.6 | 49.6 | 55.8 | 43.1 | 41.1 | 41.2 | 83.1 | 77.8 | 68.3 | 50.0 | 50.0 | 57.9 | 59.7 | 56.6 | 50.4 |
| | QAE$_{hyb}$, $\alpha=0.45$, $\beta=0.5$ | 61.6 | 58.0 | 46.8 | 64.9 | 64.9 | 70.1 | 51.2 | 48.9 | 49.0 | 85.3 | 80.5 | 69.9 | 65.6 | 65.6 | 73.4 | 70.6 | 67.2 | 61.2 |
| bce-embedding-base-v1 | - | 74.4 | 69.2 | 56.0 | 77.3 | 77.3 | 81.5 | 47.6 | 45.5 | 46.3 | 83.1 | 78.2 | 68.5 | 71.0 | 71.0 | 78.0 | 67.9 | 64.6 | 57.8 |
| | QAE$_{emb}$, $\alpha=0.3$ | 76.5 | 71.5 | 57.6 | 77.3 | 77.3 | 81.5 | 50.3 | 48.2 | 49.2 | 84.2 | 79.3 | 68.5 | 71.4 | 71.4 | 78.2 | 69.5 | 65.6 | 59.3 |
| text2vec-base-multilingual | - | 54.0 | 50.9 | 37.2 | 41.2 | 41.2 | 46.3 | 38.4 | 36.8 | 35.7 | 51.3 | 49.1 | 38.2 | 52.9 | 52.9 | 60.9 | 42.0 | 40.0 | 34.0 |
| | QAE$_{hyb}$, $\alpha=0.6$, $\beta=0.5$ | 66.7 | 63.1 | 47.9 | 56.5 | 56.5 | 61.4 | 43.7 | 42.1 | 41.7 | 72.0 | 69.0 | 53.1 | 68.0 | 68.0 | 74.8 | 57.4 | 55.3 | 47.5 |
| dpr-multiset-base | - | 77.5 | 72.5 | 60.5 | 59.4 | 59.4 | 65 | 59.7 | 57.7 | 57.5 | 82.7 | 77.3 | 69.8 | 55.9 | 55.9 | 65.1 | 70.8 | 68.2 | 60.6 |
| | QAE$_{emb}$, $\alpha=0.45$ | 82.3 | 77.4 | 64.5 | 64.3 | 64.3 | 69.3 | 60.3 | 58.1 | 58.4 | 87.1 | 82 | 72.2 | 67.7 | 67.7 | 75.3 | 74.5 | 71.8 | 64.9 |
| dpr-single-nq-base | - | 77.7 | 72.9 | 60.9 | 60.4 | 60.4 | 66.2 | 57.4 | 55.3 | 55.6 | 75.5 | 70.9 | 63.1 | 61.6 | 61.6 | 70.2 | 65.7 | 62.1 | 56.2 |
| | QAE$_{emb}$, $\alpha=0.45$ | 81.9 | 76.9 | 63.4 | 66.5 | 66.5 | 71 | 59 | 56.6 | 57.3 | 81.7 | 77.1 | 67.6 | 68.5 | 68.5 | 75.7 | 70.4 | 66.8 | 60.4 |
| **Ablation** | | | | | | | | | | | | | | | | | | | |
| bge-large-en-v1.5 | - | 87.1 | 82.9 | 68.5 | 76.3 | 76.3 | 80.4 | 58.0 | 56.0 | 56.8 | 93.4 | 89.7 | 83.6 | 75.4 | 75.4 | 81.3 | 78.5 | 74.6 | 68.1 |
| | QAE$_{emb}$, $\alpha=0.3$ | 87.4 | 83.1 | 68.5 | 78.1 | 78.1 | 81.9 | 59.4 | 56.9 | 57.8 | 93.7 | 90.0 | 83.4 | 74.8 | 74.8 | 80.8 | 79.6 | 76.2 | 70.0 |
| | QAE$_{emb}$, $\beta=0.5$ | 88.1 | 83.9 | 69.2 | 77.8 | 77.8 | 81.9 | 58.3 | 55.9 | 57.2 | 94.3 | 90.7 | 83.9 | 73.0 | 73.0 | 79.5 | 80.2 | 76.8 | 70.5 |
| | QAE$_{hyb}$, $\alpha=0.15$, $\beta=0.5$ | 88.2 | 83.8 | 69.5 | 78.4 | 78.4 | 82.5 | 59.4 | 56.8 | 58.2 | 94.2 | 90.7 | 83.6 | 73.9 | 73.9 | 80.2 | 80.3 | 77.2 | 70.8 |
| multilingual-e5-large | - | 86.0 | 81.4 | 66.8 | 86.4 | 86.4 | 89.3 | 39.4 | 38.0 | 38.7 | 95.3 | 92.2 | 85.7 | 73.7 | 73.7 | 80.3 | 76.9 | 73.6 | 66.6 |
| | QAE$_{emb}$, $\alpha=0.3$ | 87.0 | 82.0 | 67.4 | 85.5 | 85.5 | 88.3 | 45.0 | 43.4 | 43.4 | 94.8 | 92.0 | 84.7 | 70.0 | 70.0 | 77.5 | 78.6 | 75.1 | 68.3 |
| | QAE$_{emb}$, $\beta=1.5$ | 86.1 | 81.2 | 67.2 | 84.6 | 84.6 | 87.8 | 45.5 | 44.1 | 43.1 | 94.9 | 91.3 | 84.2 | 69.8 | 69.8 | 77.3 | 79.6 | 77.1 | 69.3 |
| | QAE$_{hyb}$, $\alpha=0.15$, $\beta=1.5$ | 86.1 | 81.4 | 67.3 | 85.0 | 85.0 | 88.1 | 46.7 | 45.6 | 44.5 | 94.4 | 90.6 | 83.7 | 70.2 | 70.2 | 77.5 | 79.4 | 76.8 | 69.2 |
| gte-base-en-v1.5 | - | 86.2 | 81.7 | 68.2 | 68.4 | 68.4 | 73.2 | 54.7 | 52.5 | 52.3 | 93.4 | 89.1 | 78.6 | 77.0 | 77.0 | 82.7 | 77.7 | 74.2 | 67.6 |
| | QAE$_{emb}$, $\alpha=0.3$ | 85.9 | 81.7 | 68.4 | 72.6 | 72.6 | 76.9 | 56.9 | 54.1 | 54.9 | 92.6 | 88.2 | 77.5 | 77.5 | 77.5 | 83.2 | 78.8 | 75.8 | 69.3 |
| | QAE$_{emb}$, $\beta=0.5$ | 85.4 | 81.0 | 67.4 | 73.1 | 73.1 | 77.5 | 55.7 | 53.3 | 54.0 | 93.3 | 89.4 | 78.0 | 76.8 | 76.8 | 82.6 | 78.7 | 75.3 | 69.2 |
| | QAE$_{hyb}$, $\alpha=0.3$, $\beta=0.5$ | 85.5 | 81.1 | 67.8 | 74.9 | 74.9 | 79.1 | 57.2 | 54.7 | 55.9 | 93.0 | 88.7 | 77.1 | 76.3 | 76.3 | 82.3 | 78.8 | 75.8 | 69.7 |
| jian-embeddings-v2-small-en | - | 82.4 | 78.7 | 63.7 | 69.7 | 69.7 | 74.3 | 54.5 | 52.2 | 52.5 | 92.9 | 87.6 | 78.5 | 65.5 | 65.5 | 73.9 | 74.7 | 72.5 | 64.7 |
| | QAE$_{emb}$, $\alpha=0.3$ | 83.0 | 78.8 | 64.1 | 71.5 | 71.5 | 76.0 | 54.8 | 52.4 | 53.1 | 91.6 | 86.7 | 76.6 | 65.4 | 65.4 | 73.8 | 76.3 | 73.7 | 66.2 |
| | QAE$_{emb}$, $\beta=0.5$ | 82.8 | 79.0 | 64.2 | 71.7 | 71.7 | 76.0 | 53.7 | 51.8 | 52.0 | 92.2 | 87.0 | 77.7 | 69.7 | 69.7 | 77.2 | 75.7 | 73.1 | 65.7 |
| | QAE$_{hyb}$, $\alpha=0.15$, $\beta=0.5$ | 83.3 | 79.3 | 64.5 | 72.7 | 72.7 | 76.8 | 54.1 | 52.0 | 52.5 | 91.7 | 86.7 | 77.1 | 68.4 | 68.4 | 76.3 | 76.0 | 73.6 | 66.0 |
| contriever | - | 78.8 | 74.4 | 61.4 | 65.1 | 65.1 | 70.0 | 51.5 | 49.1 | 50.6 | 89.0 | 84.0 | 74.4 | 55.8 | 55.8 | 64.7 | 71.3 | 68.2 | 61.9 |
| | QAE$_{emb}$, $\alpha=0.45$ | 84.0 | 79.5 | 66.2 | 75.1 | 75.1 | 79.5 | 55.9 | 53.8 | 55.3 | 89.9 | 85.5 | 75.5 | 67.3 | 67.3 | 74.8 | 76.4 | 73.6 | 67.9 |
| | QAE$_{emb}$, $\beta=1.25$ | 79.8 | 75.6 | 61.7 | 64.5 | 64.5 | 69.5 | 50.1 | 48.1 | 48.4 | 91.6 | 86.5 | 77.0 | 60.6 | 60.6 | 69.8 | 70.3 | 67.1 | 60.0 |
| | QAE$_{hyb}$, $\alpha=0.45$, $\beta=0.5$ | 84.1 | 79.5 | 66.1 | 72.6 | 72.6 | 77.3 | 55.3 | 52.9 | 55.0 | 89.7 | 84.6 | 75.4 | 66.7 | 66.7 | 74.2 | 75.6 | 72.1 | 66.5 |
| mcontriever | - | 52.4 | 48.2 | 38.5 | 49.6 | 49.6 | 55.8 | 43.1 | 41.1 | 41.2 | 83.1 | 77.8 | 68.3 | 50.0 | 50.0 | 57.9 | 59.7 | 56.6 | 50.4 |
| | QAE$_{emb}$, $\alpha=0.45$ | 62.3 | 58.8 | 47.2 | 63.8 | 63.8 | 69.3 | 49.4 | 47.4 | 48.2 | 85.7 | 80.9 | 70.7 | 65.0 | 65.0 | 73.2 | 70.2 | 66.4 | 61.2 |
| | QAE$_{emb}$, $\beta=0.75$ | 54.3 | 50.5 | 39.0 | 54.0 | 54.0 | 59.2 | 42.4 | 40.2 | 40.2 | 86.3 | 80.9 | 70.8 | 58.7 | 58.7 | 67.1 | 62.1 | 58.7 | 52.6 |
| | QAE$_{hyb}$, $\alpha=0.45$, $\beta=0.5$ | 61.6 | 58.0 | 46.8 | 64.9 | 64.9 | 70.1 | 51.2 | 48.9 | 49.0 | 85.3 | 80.5 | 69.9 | 65.6 | 65.6 | 73.4 | 70.6 | 67.2 | 61.2 |
| bce-embedding-base-v1 | - | 74.4 | 69.2 | 56.0 | 77.3 | 77.3 | 81.5 | 47.6 | 45.5 | 46.3 | 83.1 | 78.2 | 68.5 | 71.0 | 71.0 | 78.0 | 67.9 | 64.6 | 57.8 |
| | QAE$_{emb}$, $\alpha=0.3$ | 76.5 | 71.5 | 57.6 | 77.3 | 77.3 | 81.5 | 50.3 | 48.2 | 49.2 | 84.2 | 79.3 | 68.5 | 71.4 | 71.4 | 78.2 | 69.5 | 65.6 | 59.3 |
| | QAE$_{emb}$, $\beta=0.5$ | 75.1 | 69.8 | 56.3 | 72.8 | 72.8 | 77.9 | 50.1 | 47.5 | 48.8 | 82.2 | 76.6 | 66.4 | 72.8 | 72.8 | 79.2 | 68.7 | 65.3 | 58.4 |
| | QAE$_{hyb}$, $\alpha=0.3$, $\beta=0.5$ | 77.0 | 72.3 | 57.8 | 73.9 | 73.9 | 78.3 | 51.6 | 49.2 | 50.3 | 82.3 | 76.9 | 66.3 | 69.9 | 69.9 | 76.9 | 69.8 | 65.8 | 59.4 |
| text2vec-base-multilingual | - | 54.0 | 50.9 | 37.2 | 41.2 | 41.2 | 46.3 | 38.4 | 36.8 | 35.7 | 51.3 | 49.1 | 38.2 | 52.9 | 52.9 | 60.9 | 42.0 | 40.0 | 34.0 |
| | QAE$_{emb}$, $\alpha=0.75$ | 66.7 | 62.8 | 47.9 | 55.0 | 55.0 | 60.0 | 43.2 | 41.6 | 41.5 | 71.9 | 68.4 | 53.3 | 69.2 | 69.2 | 75.7 | 58.0 | 55.7 | 48.1 |
| | QAE$_{emb}$, $\beta=0.5$ | 56.8 | 54.0 | 40.5 | 46.8 | 46.8 | 52.0 | 39.1 | 37.9 | 36.7 | 61.5 | 59.0 | 45.8 | 57.8 | 57.8 | 65.5 | 47.2 | 45.2 | 38.8 |
| | QAE$_{hyb}$, $\alpha=0.6$, $\beta=0.5$ | 66.7 | 63.1 | 47.9 | 56.5 | 56.5 | 61.4 | 43.7 | 42.1 | 41.7 | 72.0 | 69.0 | 53.1 | 68.0 | 68.0 | 74.8 | 57.4 | 55.3 | 47.5 |

Table 11: Comprehensive retrieval performance on the latest datasets FIGNEWS and CRUD-RAG (**Top-k = 10**). Higher is better, with the best one bolded. Hyperparameters including QAEncoder variants and weight terms $\alpha$, $\beta$ are optimized simultaneously for six latest datasets. '-' denotes default or null values.

| Model | Param | FIGNEWS(English) | | FIGNEWS(Arabic) | | CRUD-RAG(Chinese) | | FIGNEWS(French) | | FIGNEWS(Hindi) | | FIGNEWS(Hebrew) | |
|---|---|---|---|---|---|---|---|---|---|---|---|---|---|
| | | MRR@10 | NDCG@10 | MRR@10 | NDCG@10 | MRR@10 | NDCG@10 | MRR@10 | NDCG@10 | MRR@10 | NDCG@10 | MRR@10 | NDCG@10 |
| bge-m3 | - | 74.4 | 78.7 | 77.8 | 80.9 | 86.9 | 89 | 73.5 | 77.4 | 58.6 | 64.4 | 78.7 | 81.4 |
| | $QAE_{ext}, \beta=1.5$ | 77.2 | 81 | 80.2 | 83.1 | 89 | 90.9 | 76.9 | 80.3 | 62.7 | 67.8 | 80 | 82.8 |
| multilingual-e5-small | - | 71 | 75.1 | 74.1 | 77.4 | 81.7 | 84.2 | 66.8 | 70.4 | 52.5 | 57.8 | 72.9 | 76.5 |
| | $QAE_{hyb}, \alpha=0.3, \beta=0.5$ | 74.6 | 78.5 | 78.9 | 81.6 | 87.5 | 89.3 | 74.2 | 77.9 | 59.6 | 64.6 | 77.5 | 80.4 |
| multilingual-e5-base | - | 74.8 | 78.1 | 72.3 | 76 | 86.1 | 88.2 | 71.2 | 75.1 | 57.8 | 62.9 | 72.6 | 75.8 |
| | $QAE_{emb}, \alpha=0.3$ | 77.6 | 81.3 | 77.2 | 80.3 | 88.5 | 90.5 | 76.7 | 80 | 61.5 | 66.5 | 77.7 | 80.5 |
| multilingual-e5-large | - | 73.9 | 77.8 | 76.7 | 80.2 | 85.8 | 88.3 | 70.6 | 74.5 | 53 | 59.2 | 73.9 | 77.4 |
| | $QAE_{hyb}, \alpha=0.15, \beta=1.25$ | 77.1 | 80.6 | 82.2 | 85.1 | 89.1 | 91.2 | 77.4 | 80.9 | 60.6 | 65.9 | 77.7 | 81 |
| gte-multilingual-base | - | 65.5 | 70.4 | 73.4 | 76.8 | 82.9 | 85.7 | 63 | 67.4 | 52.4 | 58.6 | 66 | 69.6 |
| | $QAE_{hyb}, \alpha=0.15, \beta=1.5$ | 75.5 | 79.5 | 76.2 | 79.1 | 85.5 | 88.2 | 66.9 | 71.6 | 56 | 61.9 | 74 | 77.6 |
| mcontriever | - | 32.9 | 36.7 | 40.3 | 44.7 | 71.8 | 76.2 | 35 | 39.3 | 27 | 31.8 | 49.5 | 54.4 |
| | $QAE_{hyb}, \alpha=0.45, \beta=1.25$ | 61.4 | 65.9 | 68.3 | 72.1 | 88.7 | 90.6 | 64.7 | 69.1 | 50.6 | 56.4 | 70.1 | 73.8 |
| bce-embedding-base-v1 | - | 59.1 | 63.8 | - | - | 76.9 | 80.5 | - | - | - | - | - | - |
| | $QAE_{hyb}, \alpha=0.3, \beta=0.5$ | 66.8 | 71.1 | - | - | 86 | 88.3 | - | - | - | - | - | - |
| text2vec-base-multilingual | - | 38.7 | 43.6 | 27.8 | 31.9 | 12.2 | 13.9 | 33.7 | 38.6 | 15.6 | 19.9 | 12.6 | 15.5 |
| | $QAE_{emb}, \alpha=0.75$ | 55.4 | 59.9 | 51.5 | 55.4 | 55.5 | 59 | 49.3 | 54.4 | 36.2 | 41.5 | 47.1 | 52.1 |
| **Ablation** | | | | | | | | | | | | | |
| bge-m3 | - | 74.4 | 78.7 | 77.8 | 80.9 | 86.9 | 89 | 73.5 | 77.4 | 58.6 | 64.4 | 78.7 | 81.4 |
| | $QAE_{emb}, \alpha=0.3$ | 76.4 | 80.5 | 80.1 | 82.9 | 88.8 | 90.7 | 75 | 78.5 | 61 | 66 | 79 | 82 |
| | $QAE_{ext}, \beta=1.5$ | 77.2 | 81 | 80.2 | 83.1 | 89 | 90.9 | 76.9 | 80.3 | 62.7 | 67.8 | 80 | 82.8 |
| | $QAE_{hyb}, \alpha=0.15, \beta=1.5$ | 77.4 | 81.1 | 80.6 | 83.4 | 89.4 | 91.2 | 76.5 | 80 | 61.7 | 66.8 | 79.8 | 82.7 |
| multilingual-e5-small | - | 71 | 75.1 | 74.1 | 77.4 | 81.7 | 84.2 | 66.8 | 70.4 | 52.5 | 57.8 | 72.9 | 76.5 |
| | $QAE_{emb}, \alpha=0.45$ | 74.7 | 78.5 | 77 | 79.8 | 87.8 | 89.8 | 73.6 | 77.1 | 58 | 63.2 | 77.1 | 80.1 |
| | $QAE_{ext}, \beta=1.0$ | 73.2 | 77.2 | 79.2 | 81.9 | 85 | 89.2 | 70.6 | 74.7 | 58.7 | 63.7 | 77.1 | 80.2 |
| | $QAE_{hyb}, \alpha=0.3, \beta=0.5$ | 74.6 | 78.5 | 78.9 | 81.6 | 87.5 | 89.3 | 74.2 | 77.9 | 59.6 | 64.6 | 77.5 | 80.4 |
| multilingual-e5-base | - | 74.8 | 78.1 | 72.3 | 76 | 86.1 | 88.2 | 71.2 | 75.1 | 57.8 | 62.9 | 72.6 | 75.8 |
| | $QAE_{emb}, \alpha=0.3$ | 77.6 | 81.3 | 77.2 | 80.3 | 88.5 | 90.5 | 76.7 | 80 | 61.5 | 66.5 | 77.7 | 80.5 |
| | $QAE_{ext}, \beta=0.75$ | 74.7 | 78.8 | 76.4 | 79.7 | 87.6 | 89.7 | 72.3 | 76.4 | 62.5 | 67.8 | 77 | 79.9 |
| | $QAE_{hyb}, \alpha=0.3, \beta=0.5$ | 76.6 | 80.4 | 77.4 | 80.6 | 89.1 | 90.9 | 75.4 | 79.1 | 62.7 | 67.7 | 77.2 | 80.2 |
| multilingual-e5-large | - | 73.9 | 77.8 | 76.7 | 80.2 | 85.8 | 88.3 | 70.6 | 74.5 | 53 | 59.2 | 73.9 | 77.4 |
| | $QAE_{emb}, \alpha=0.45$ | 77.9 | 81.4 | 79.8 | 83 | 89.8 | 91.5 | 77 | 80 | 58.3 | 63.5 | 78.1 | 81.2 |
| | $QAE_{ext}, \beta=1.5$ | 75.6 | 79.2 | 80.9 | 84.1 | 88.3 | 90.5 | 76 | 79.4 | 60 | 65.5 | 77 | 80.2 |
| | $QAE_{hyb}, \alpha=0.15, \beta=1.25$ | 77.1 | 80.6 | 82.2 | 85.1 | 89.1 | 91.2 | 77.4 | 80.9 | 60.6 | 65.9 | 77.7 | 81 |
| gte-multilingual-base | - | 65.5 | 70.4 | 73.4 | 76.8 | 82.9 | 85.7 | 63 | 67.4 | 52.4 | 58.6 | 66 | 69.6 |
| | $QAE_{emb}, \alpha=0.45$ | 69.1 | 73.6 | 76.9 | 79.6 | 87.3 | 89.4 | 67.1 | 71.2 | 53 | 58.5 | 72.4 | 75.8 |
| | $QAE_{ext}, \beta=1.5$ | 75.7 | 79.7 | 75.9 | 78.9 | 84.5 | 87.3 | 66.1 | 70.7 | 56.8 | 62.5 | 72.8 | 76.5 |
| | $QAE_{hyb}, \alpha=0.15, \beta=1.5$ | 75.5 | 79.5 | 76.2 | 79.1 | 85.5 | 88.2 | 66.9 | 71.6 | 56 | 61.9 | 74 | 77.6 |
| mcontriever | - | 32.9 | 36.7 | 40.3 | 44.7 | 71.8 | 76.2 | 35 | 39.3 | 27 | 31.8 | 49.5 | 54.4 |
| | $QAE_{emb}, \alpha=0.6$ | 58.8 | 64.2 | 67.2 | 71.3 | 88.3 | 90.2 | 62.7 | 66.9 | 50.4 | 55.7 | 69.7 | 73.2 |
| | $QAE_{ext}, \beta=1.5$ | 49.2 | 54.2 | 59.5 | 63.9 | 80 | 84.3 | 54.1 | 58.9 | 45.1 | 50.2 | 60.8 | 64.6 |
| | $QAE_{hyb}, \alpha=0.45, \beta=1.25$ | 61.4 | 65.9 | 68.3 | 72.1 | 88.7 | 90.6 | 64.7 | 69.1 | 50.6 | 56.4 | 70.1 | 73.8 |
| bce-embedding-base-v1 | - | 59.1 | 63.8 | - | - | 76.9 | 80.5 | - | - | - | - | - | - |
| | $QAE_{emb}, \alpha=0.45$ | 66.8 | 71.3 | - | - | 85.3 | 87.6 | - | - | - | - | - | - |
| | $QAE_{ext}, \beta=1.5$ | 64.3 | 68.7 | - | - | 82.3 | 85.1 | - | - | - | - | - | - |
| | $QAE_{hyb}, \alpha=0.3, \beta=0.5$ | 66.8 | 71.1 | - | - | 86 | 88.3 | - | - | - | - | - | - |
| text2vec-base-multilingual | - | 38.7 | 43.6 | 27.8 | 31.9 | 12.2 | 13.9 | 33.7 | 38.6 | 15.6 | 19.9 | 12.6 | 15.5 |
| | $QAE_{emb}, \alpha=0.75$ | 55.4 | 59.9 | 51.5 | 55.4 | 55.5 | 59 | 49.3 | 54.4 | 36.2 | 41.5 | 47.1 | 52.1 |
| | $QAE_{ext}, \beta=1.5$ | 46.2 | 50.8 | 35.5 | 39.4 | 21.3 | 23.6 | 43.3 | 47.7 | 21.3 | 25.2 | 18.8 | 22.1 |
| | $QAE_{hyb}, \alpha=0.75, \beta=0.5$ | 55.1 | 59.7 | 50.4 | 54.4 | 57.8 | 60.7 | 49.1 | 53.8 | 34.5 | 39.6 | 47.1 | 51.5 |

Table 12: Complete performance comparison of QAEncoder variants on latest datasets FIGNEWS and CRUD-RAG (**Top-k = 10**). Higher is better, with the best one bolded. Hyperparameters are optimized simultaneously across the six latest datasets. $n$ indicates the number of predicted queries in $QA_{naive}$.

| Model | Param | FIGNEWS(English) | | FIGNEWS(Arabic) | | CRUD-RAG(Chinese) | | FIGNEWS(French) | | FIGNEWS(Hindi) | | FIGNEWS(Hebrew) | |
|---|---|---|---|---|---|---|---|---|---|---|---|---|---|
| | | MRR@10 | NDCG@10 | MRR@10 | NDCG@10 | MRR@10 | NDCG@10 | MRR@10 | NDCG@10 | MRR@10 | NDCG@10 | MRR@10 | NDCG@10 |
| bg3-m3 | $QAE_{emb}, \alpha=0.3$ | 76.4 | 80.5 | 80.1 | 82.9 | 88.8 | 90.7 | 75 | 78.5 | 61 | 66 | 79 | 82 |
| | $QAE_{ext}, \beta=1.5$ | 77.2 | 81 | 80.2 | 83.1 | 89 | 90.9 | 76.9 | 80.3 | 62.7 | 67.8 | 80 | 82.8 |
| | $QAE_{hyb}, \alpha=0.3, \beta=1.5$ | 77.4 | 81.1 | 80.6 | 83.4 | 89.4 | 91.2 | 76.5 | 80 | 61.7 | 66.6 | 79.8 | 82.7 |
| | $QA_{naive}, n=10$ | 76.9 | 79.9 | 77.1 | 79.7 | 86 | 88 | 71.9 | 74.4 | 62.3 | 66.1 | 68.1 | 71.7 |
| multilingual-e5-large | $QAE_{emb}, \alpha=0.45$ | 77.9 | 81.4 | 79.8 | 83 | 89.8 | 91.5 | 77 | 80 | 58.3 | 63.5 | 78.1 | 81.2 |
| | $QAE_{ext}, \beta=1.5$ | 75.6 | 79.2 | 80.9 | 84.1 | 88.3 | 90.5 | 76 | 79.4 | 60 | 65.5 | 77 | 80.2 |
| | $QAE_{hyb}, \alpha=0.15, \beta=1.25$ | 77.1 | 80.6 | 82.2 | 85.1 | 89.1 | 91.2 | 77.4 | 80.9 | 60.6 | 65.9 | 77.7 | 81 |
| | $QA_{naive}, n=10$ | 77.5 | 80.3 | 76.5 | 79.4 | 85.1 | 87.3 | 70.5 | 73.5 | 61.5 | 65.6 | 69.4 | 72.2 |

Table 13: The table illustrates a comprehensive performance comparison of QAEncoder against training-based and document-centric methods on the latest datasets: FIGNEWS and CRUD-RAG (**Top-k = 10**). Higher is better, with the best one bolded. Hyperparameters $\alpha$, $\beta$ are optimized simultaneously across the six latest datasets. $n$ denotes the number of pseudo-documents in HyDE. '-' indicates default or null values.

| Model | Param | FIGNEWS(English) | | FIGNEWS(Arabic) | | CRUD-RAG(Chinese) | | FIGNEWS(French) | | FIGNEWS(Hindi) | | FIGNEWS(Hebrew) | |
|---|---|---|---|---|---|---|---|---|---|---|---|---|---|
| | | MRR@10 | NDCG@10 | MRR@10 | NDCG@10 | MRR@10 | NDCG@10 | MRR@10 | NDCG@10 | MRR@10 | NDCG@10 | MRR@10 | NDCG@10 |
| mcontriever-msmarco | - | 66.1 | 70.6 | 70.2 | 73.7 | 85.1 | 87.5 | 66.4 | 70.3 | 48.7 | 53.9 | 69.4 | 72.9 |
| | $QAE_{hyb}, \alpha=0.3, \beta=0.75$ | 72.3 | 76.8 | 77.3 | 80.5 | 88.6 | 90.7 | 74.4 | 78 | 59 | 64.1 | 76.5 | 79.8 |
| multilingual-e5-large-instruct | - | 67 | 71.4 | 75 | 78.2 | 79.9 | 82.8 | 66 | 70.9 | 48.2 | 53.8 | 69.6 | 73.3 |
| | $QAE_{hyb}, \alpha=0.15, \beta=1.5$ | 75.6 | 79.8 | 80.8 | 83.7 | 88.9 | 90.6 | 75.8 | 79.5 | 59.3 | 64.6 | 79.4 | 82.3 |
| mcontriever | - | 32.9 | 36.7 | 40.3 | 44.7 | 71.8 | 76.2 | 35 | 39.3 | 27 | 31.8 | 49.5 | 54.4 |
| | HyDE, n=8 | 25 | 27.9 | 35.7 | 41.9 | 70.9 | 74.7 | 25.8 | 28.9 | 11.8 | 14.9 | 42.1 | 47.8 |
| multilingual-e5-large | - | 73.9 | 77.8 | 76.7 | 80.2 | 85.8 | 88.3 | 70.6 | 74.5 | 53 | 59.2 | 73.9 | 77.4 |
| | HyDE, n=8 | 63.6 | 68.3 | 68.3 | 74.1 | 81.6 | 84.1 | 58.4 | 63.2 | 45.6 | 51.3 | 65.3 | 69.6 |

