# OpenReview forum: "QAEncoder: Towards Aligned Representation Learning in Question Answering System"
_ICLR.cc/2025/Conference — ICLR 2025 Conference Withdrawn Submission_

### Official Review · Reviewer_zDcY · 2024-10-22

**Soundness:** 2
**Presentation:** 2
**Contribution:** 2
**Rating:** 3
**Confidence:** 4

**Summary:**

The authors propose QAEncoder, a training-free approach that utilizes a large language model to generate a pseudo query set from each document in the corpus. They introduce various methods to augment the document representations embedded by the retriever (referred to as "document fingerprints" in the paper) based on the generated queries. The approach is evaluated on QA and cross-lingual retrieval benchmarks, comparing the performance of retrieval models augmented with this method to the original ones.

**Strengths:**

1. The authors propose several methods for augmenting document embeddings, which could be valuable. Some of their findings are interesting and may provide insights into more effective techniques for representation augmentation.

2. The authors also present various hypotheses and theories, thoughtfully connecting them in a manner that could serve as a useful reference for future research and exploration.

**Weaknesses:**

1. The evaluation process has several issues:

    - The evaluation metric does not align with the well-established metrics used in prior research, such as retrieval accuracy for NQ and SQuAD, or MRR@10 and NDCG@10 for MS MARCO. Changing it to MRR@8 seems unnecessary, and the authors should either justify this choice or also provide results using the well-established metrics.

    - Most of the retrieval baselines are outdated or unsupervised (e.g., Contriever) or are cross-lingual retrievers. It would strengthen the evaluation if more general neural retrieval models, such as DPR, were included. Additionally, some advanced models are evaluated only in their multilingual versions, and it’s not entirely clear how this focus aligns with the authors' overall motivation.

2. The improvements are minor, considering that the benchmarked retrievers are either weak due to limited training data in their multilingual versions or are outdated, unsupervised models. This is particularly concerning given that the authors use the most recent GPT-4-mini for their enhancements. From another perspective, the proposed solution does not appear to be cost-effective.

3. Some concepts are overly wrapped and lack consensus in the field. For example, the "conical distribution hypothesis" occupies a significant amount of space but does not appear relevant or convincing. Although it may not be false, it is not currently reflected or validated during the retriever’s pre-training or tuning process. Instead, the validity of this hypothesis and the proposed solution might heavily depend on the prompt used to generate the pseudo query and how well it aligns with downstream tasks. Additionally, the term "fingerprint" could benefit from further clarification, as it seems to refer to augmenting document embeddings with pseudo queries. If this interpretation is incorrect, additional explanation would help, as the term might be perceived as vague, making the concept harder to grasp. Demonstrating the solution in a simple, direct manner could enhance clarity.

**Questions:**

1. Could the authors please provide more clarity on the concept of the "document-query gap" discussed in this paper and how the proposed method addresses it? Is this gap intended to be reflected through performance improvements, or is it an issue that might not be fully captured by existing benchmarks?

2. I am asking Q1 because I am a bit unclear about the motivation behind this training-free augmentation approach. In what scenario is it necessary, and does it address an issue that benchmarks might not fully capture? While the idea is interesting, the improvement seems somewhat inconsistent and not particularly significant, especially considering the high cost involved (e.g., using GPT-4 to generate a pseudo query for each document).

3. Following Q2, could you provide the number of generated tokens or text chunks required to complete the tasks? Given a similar cost, would it be more effective to label a training set and conduct in-domain training instead?

4. Could you please share the reasoning behind focusing on the cross-lingual scenario rather than comparing the method using other well-established retrieval benchmark?

**Details Of Ethics Concerns:**

To the best of my knowledge, there are no ethical concerns associated with this work.

---

> ### Author Response · Authors · 2024-11-21
> **Author Response [1/3]**
>
> Dear Reviewer zDcY:
>
> Thank you so much for your comments and efforts on our research work. Before addressing your questions individually, let us clarify several important facts.
>
> ## F1. (Concept of Document-Query Gap)
> The term **\"document-query gap\"** refers to the differences between documents and queries across **lexical, syntactic, semantic, and content dimensions**. E.g., documents are often lengthy and declarative, whereas queries tend to be brief and interrogative. This inherent gap, akin to the modality gaps observed between text and image (CLIP) or text and code, hinders representation alignment and reduces cosine similarities for matching. This gap manifests as **a decreased retrieval performance in benchmarks**.
>
> ## F2. (Addressing Document-Query Gap)
> Previous approaches include generating pseudo-documents at inference time (document-centric methods) or adopting QA datasets for domain adaptation (training-based methods). QAEncoder bridges this gap mainly by **estimating the cluster center of potential queries in embedding space as a surrogate for the document embedding**.
>
> **The classical benchmarks** in Table 1 have been extensively domain-adapted by SOTA models, thus **this gap is alleviated and cannot be fully captured**. When benchmarked **on the latest datasets** in Table 2 and Table 4, **this gap is unveiled again**. Previous works suffer hallucination or limited generalization, while QAEncoder performs well. Moreover, integrating training-based approaches with QAEncoder could lead to more improvements.
>
> ## F3. (Cost-Effectiveness)
> Briefly, QAEncoder is cost-effective and efficient.
>
> Firstly, the cost mainly comes from the query generation process. As will be responded in Q3, **10 queries with an average length of 6-15 tokens** are typically sufficient for ordinary documents.
>
> Secondly, **GPT-4o-mini is currently the cheapest generative model by OpenAI**, priced at \\$0.075 per million input tokens and \$0.300 per million output tokens \[9\]. The most recent Qwen2.5-Turbo by Alibaba is even 3.6 times cheaper than GPT-4o-mini \[10\]. Hence, about **0.1 million documents can be processed within 1 dollar** via API call ($\frac{1\mathrm{M}\*3.6}{0.3*(10*10)} \approx 0.1\mathrm{M}$). The prices will continually decrease as AI and small language models develop.
>
> Thirdly, the initial query-centric work for sparse retrievers, DocT5Query \[11\], demonstrates **T5-base with 0.2B parameters is sufficient for query generation, while no improvement are gained with larger models**. We also confirmed Qwen2.5-0.5B-Instruct as a good query generator in our business implementation. We choose GPT-4o-mini just given its **out-of-the-box and comprehensively multilingual support for academic research**.
>
> Finally, in our data flow settings (batch size=1), 2~3 documents can be processed per second with Qwen2.5-0.5B-Instruct + vLLM + BF16 on a single NVIDIA A100 80GB \[14\]. For batch processing with higher GPU utility, DocT5Query reports sampling 5 queries per document for **8.8M MSMARCO documents requires approximately 40 hours on a single Google TPU v3, costing only \$96 USD** (40 hours × \$2.40 USD/hour) in 2019 \[11\].
>
> ## F4. (Experiment Design and Results)
> As stated in the paper, we employed **several SOTA baselines with extensive training and fine-tuning on classical or well-established datasets**. For instance, BGE and E5 models have been fine-tuned on all of these datasets \[1,7\], which should reach performance limits on these datasets.
>
> For the most advanced models, on the one hand, **the results on classical datasets primarily reflect the in-domain or multi-domain adaptation capabilities**. Experiments show that although this adaptation can alleviate the document-query gap, it suffers from **common issues in multi-task learning like negative transfer and task imbalance** despite classical QA datasets are in similar domains \[5\]. **E.g., E5 performs poorly on its fine-tuned dataset ELI5, and MRR increases from 39.0 to 46.4 with QAEncoder equipped.**
>
> On the other hand, in scenarios such as search engines and news QA, large volumes of new data constantly emerge and are indexed. **The retrieval performance on rapidly and lifelong updated datasets is of great interest**, where domain adaptation is challenged and not effective anymore (as far as we know, existing domain-adaption methods are mainly focused and tested on a single domain). **Hence, we propose to conduct tests on the latest datasets without any domain-adaption.** To confirm the multilingual applicability of QAEncoder simultaneously, we considered multilingual datasets in our selection of the latest datasets. Experiments on the latest datasets show the significant and consistent improvements by QAEncoder on **all tested models, datasets and languages**. To address concerns about the weakness of multilingual models, **we have appended results from monolingual models and bilingual models in the Appendix**.

---

> ### Author Response · Authors · 2024-11-21
> **Author Response [2/3]**
>
> ## Response to W1
> > **(Metrics and Baselines)** The evaluation process has issues: it uses an unaligned metric, MRR@8, over standard ones like accuracy or MRR@10/NDCG@10, lacking justification. Most baselines are weak or outdated, like Contriever, or cross-lingual. Including general neural models, like DPR, and clarifying the focus on multilingual models would strengthen the evaluation.
>
> For metrics, accuracy, recall, and precision are set-based measures that fail to capture the ranking of retrieval results. **Instead, MRR, MAP and NDCG reflect both the relevance and relative order**, and are increasingly popular. For top-k parameter, the choice between 8 and 10 does not impact conclusions. We appreciate your valuable feedback and **have appended the results with top-k=10 in the Appendix**.
>
> For baselines, we clarify that, we experiment with many strong baselines.
>
> For sparse retrievers, we included them to validate the superiority of dense retrievers and emphasize the significance of query-centric methods for dense retrieval.
>
> For dense retrievers, we conduct extensive experiments on **fourteen embedding models including the state-of-the-art BGE (BAAI, 2024) \[13\], E5 (Microsoft, 2023) \[2\], GTE (Alibaba, 2024) \[4,8\], Jina (Jina AI, 2023) \[3\]**, all as strong baselines. DPR (Facebook, 2020a) \[6\], the seminal dense retriever, supervisedly fine-tunes BERT on QA datasets without additional pre-training, and thus still struggles with cross-domain scenarios. Contriever (Facebook, 2021b) \[12\], the unsupervisedly pre-trained version, is nearly on par with DPR tested on DPR's datasets. After fine-tuned on MSMARCO dataset, contriever-msmarco beats DPR on BEIR benchmark significantly.
>
> Regarding multilingual models, we adopted them **to provide more valid results** on multilingual datasets and demonstrating the applicability of our approach across various languages. **To address concerns on the weakness of multilingual models, we've included DPR, monolingual model and bilingual model results in the Appendix**. Thank you for your careful reading and valuable feedback!
>
> ## Response to W2
> > **(Improvements and Cost)** The improvements are minor, as retrievers are either undertrained in multilingual versions or outdated unsupervised models. And the solution does not appear to be cost-effective.
>
> For the improvement concerns, as clarified in W1, **we tested many strong baselines and achieved significant improvements on the latest datasets**. The results of monolingual and bilingual models are included in the appendix to address concerns about the weakness of multilingual models.
>
> For the cost concerns, our method is indeed **cost-effective**, **as elaborated in F3**.
>
> ## Response to W3
> > **(Concepts Validation and Presentation)** Some concepts are overly wrapped and lack consensus. The conical distribution hypothesis does not appear relevant or convincing. Although it may not be false, it is not currently reflected or validated during pre-training or tuning process. Instead, the validity of this hypothesis and the proposed solution might depend on the query generation prompt. Additionally, the term fingerprint could benefit from further clarification.
>
> Thank you for feedbacks. **We believe the proposed concepts are intuitive and useful. Additionally, other reviewers consistently praised our presentation.** For instance, visualizations of architecture and hypotheses make them more easy to understand.
>
> Firstly, our hypothesis is empirically true. Figure 3 intuitively presents verification and illustration. **This verification has been conducted with various embedding models, languages, documents and queries**. The experiment is simple and easily reproducible via our provided repo: https://anonymous.4open.science/r/ICLR-QAEncoder. Besides, we use **a standard query generation prompt from llamaindex without specific modifications.**
>
> Secondly, this hypothesis is significantly relevant. Because it captures the geometric relationship between documents and potential queries in the embedding space. **It ensures the rationality and effectiveness of $QAE_{\text{base}}$** fundamentally.
>
> Thirdly, **our hypothesis is novel and thus lacks common knowledge**. The causes and dynamics underlying this hypothesis are interesting but complicated, and their exploration falls outside the scope of this paper. Though our hypothesis is highly simplified, we believe it can **inspire and contribute to better model design in this field**.
>
> Document fingerprints intuitively refer to documents' characteristic information. The initial form of QAEncoder, $QAE_{\text{base}}$, faces the document distinguishability issue, as too query-like. Hence, document fingerprints are integrated in several ways. E.g., $QAE_{emb}$ and $QAE_{txt}$ extend $QAE_{\text{base}}$ with fingerprints in the embedding and textual spaces respectively.
>
> Let us know if any part requires more clarification.

---

> > ### Author Response · Authors · 2024-11-21
> > **Author Response [3/3]**
> >
> > ## Response to Q1
> > > **(Document-Query Gap)** Could the authors provide more clarity on the concept of the \"document-query gap\"  and how the proposed method addresses it? Is this gap intended to be reflected through performance improvements, or is it an issue that might not be fully captured by existing benchmarks?
> >
> > Yes, **existing classical benchmarks cannot fully capture this gap due to extensively domain-adapted by SOTA models**. Please refer to F1 (Concept of Document-Query Gap), F2 (Addressing Document-Query Gap) and F4 (Experiment Design and Results) for details. Thanks!
> >
> > ## Response to Q2
> > > **(Motivation of Training-Free Approach)** I am a bit unclear about the motivation behind this training-free augmentation approach. In what scenario is it necessary, and does it address an issue that benchmarks might not fully capture? While the idea is interesting, the improvement seems somewhat inconsistent and not particularly significant, especially considering the high cost involved (e.g., using GPT-4 to generate pseudo queries).
> >
> > Briefly, (1) training-free augmentation is desirable and indispensable in **large-scale real-world applications with rapid and lifelong knowledge updates**; (2) classical benchmarks cannot fully capture this gap due to extensively domain-adapted by SOTA models, **while the latest datasets unveil this gap** and issues like limited generalization of training-based methods; (3) the improvements on these scenarios are **consistently significant while remains cost-effective**.
> >
> > Please refer to F2 (Addressing Document-Query Gap) and F4 (Experiment Design and Results) for (1) and (2) details; refer to F3 (Cost-Effectiveness) and F4 (Experiment Design and Results) for (3) details. Thanks!
> >
> > ## Response to Q3
> > > **(Cost-Effectiveness)** Could you provide the number of generated tokens or text chunks required? Given a similar cost, would it be more effective to label a training set and conduct in-domain training?
> >
> > As implemented, the number of generated queries scales linearly with the document length. **For a chunk size of approximately 1024, generating 10 queries (6~15 tokens each) is generally adequate**. As shown in Figure 6, for documents exceeding 150 words in the MSMARCO dataset, the cluster center estimation with 10 queries achieves a similarity score of 0.96 compared with 80 queries. For MSMARCO (\~56 words/doc) and NQ (~78 words/doc), about 6 and 8 queries are generated per document respectively. Our method is **cost-effective and acceptable**, please see F3 (Cost-Effectiveness) for more cost details.
> >
> > While in-domain training enhances performance within a specific domain, **existing domain-adaptation methods mainly focus on single domains**. In pursuit of **more dynamic and diverse scenarios**, it is challenged by practical issues in continual training. The training-free nature makes QAEncoder indispensable. Please see F4 (Experiment Design and Results) for more details. Thanks!
> >
> > ## Response to Q4
> > > **(Benchmark Consideration)** Could you please share the reasoning behind focusing on the cross-lingual scenario rather than comparing the method using other well-established benchmark?
> >
> > We clarify that our focus is not specifically on multilingual or crosslingual scenarios. Instead, **we focus on the latest datasets**, while multilingual datasets are used to **validate multilingual applicability simultaneously**.
> >
> > Though the well-established benchmarks are included in our paper. As responded in Q1, for extensively fine-tuned SOTA models, **the classical or well-established benchmarks cannot fully capture the document-query gap and other issues like limited generalization. The latest datasets help unveil these issues again**, and confirm the significance of QAEncoder. Please see F4 (Experiment Design and Results) for details. Thanks!
> >
> > Finally, thank you again for your detailed comments and valuable feedback. We acknowledge the room for improvements. If you have any further questions or comments, please let us know at any time.
> >
> > ## Reference
> > [1] https://huggingface.co/datasets/Shitao/bge-m3-data
> >
> > [2] Multilingual e5 text embeddings: A technical report
> >
> > [3] Jina embeddings 2: 8192-token general-purpose text embeddings for long documents
> >
> > [4] Towards general text embeddings with multi-stage contrastive learning
> >
> > [5] Domain adaptation and multi-domain adaptation for neural machine translation: A survey
> >
> > [6] Dense passage retrieval for open-domain question answering
> >
> > [7] https://huggingface.co/intfloat/multilingual-e5-large
> >
> > [8] mGTE: Generalized Long-Context Text Representation and Reranking Models for Multilingual Text Retrieval
> >
> > [9] https://openai.com/api/pricing
> >
> > [10] https://qwen2.org/qwen2-5-turbo
> >
> > [11] From doc2query to docTTTTTquery
> >
> > [12] Unsupervised dense information retrieval with contrastive learning
> >
> > [13] C-pack: Packed resources for general chinese embeddings
> >
> > [14] https://qwen.readthedocs.io/en/latest/benchmark/speed_benchmark.html

---

> > > ### Comment · Reviewer_zDcY · 2024-11-25
> > >
> > > Thank you for your detailed response. Given the tight timeline, I have just a quick question in W1.
> > >
> > > ### Re W1
> > >
> > > The MSMARCO MRR@10 scores in Table 10 seem quite different from those in other papers [1]. Could you clarify if this is due to differences in setup or evaluation methods?
> > >
> > > ### Re W2
> > > > For dense retrievers, we conduct extensive experiments on fourteen embedding models including the state-of-the-art BGE (BAAI, 2024) [13], E5 (Microsoft, 2023) [2], GTE (Alibaba, 2024) [4,8], Jina (Jina AI, 2023) [3]
> > >
> > > Most of the models you compare against are multilingual versions (e.g., multilingual-E5 rather than E5) on multilingual tasks. For instance, multilingual-E5 is tuned on 500k synthetic data generated by GPT-3.5/4. While both could be referred to as E5, they are different. Additionally, your method utilizes synthetic data from GPT-4o at a million-scale, which likely benefits from stronger multilingual capabilities. Given this, My major concern remain as **it is hard to justify whether the improvement comes from the proposed approach or the multilingual synthetic data from GPT-4o**. I still believe it would be more justified and straightforward to conduct an evaluation using mainstream IR benchmarks (e.g., the entire BEIR) and non-multilingual SOTA retrieval systems (e.g., E5 instead of multilingual-E5) to enable a fairer comparison.
> > >
> > > ### Re W3
> > >
> > > > We believe the proposed concepts are intuitive and useful. Additionally, other reviewers consistently praised our presentation.
> > >
> > > I appreciate your clarification, and I noticed that other reviewers raised similar concerns. Reviewer coVe questions the relevance of your hypothesis to your method, and Reviewer sfJJ finds the concept of QAE fingerprints confusing. I personally appreciate simple and direct ideas, and I find the core idea of your work straightforward. However, the hypothesis seems less connected to the method, and the introduction of overly complex concepts makes it less accessible. I suggest removing the overly wrapped concepts and keeping the presentation straightforward.
> > >
> > > [1] Shen, Tao, et al. "Lexmae: Lexicon-bottlenecked pretraining for large-scale retrieval."

---

> ### Author Response · Authors · 2024-11-24
> **Official Comment by Authors**
>
> Dear reviewer zDcY:
>
> Thanks again for your valuable comments. We hope that our response could address your concerns. Please do not hesitate to ask if you have any questions about our manuscript or response.

---

> ### Author Response · Authors · 2024-11-25
> **Author Response (2nd Round)**
>
> Dear Reviewer zDcY:
>
> Thank you again for your comments and suggestions. Let us further clarify some facts and respond to your concerns.
>
> ### Response to W1
> > Could you clarify if this is due to differences in setup or evaluation methods?
>
> Yes, this is due to the experimental setup.
>
> After careful examination, the discrepancies should arise from the dataset version. In our experiments, we adopted the MSMARCO v1.1 dataset, which is roughly a 1/10 subset of the MSMARCO v2.1 dataset [1].
>
> But we mention that, for comparative analysis, the discrepancies of absolute scores should not compromise the evaluation conclusion of relative performance. Besides, to capture the document-query gap, our research focuses on the latest datasets rather than classical ones (See F4).
>
> We appreciate you pointing out this aspect and will supplement MSMARCO v2.1 in the revision.
>
> ### Response to W2
> > It is hard to justify whether the improvement comes from the proposed approach or the multilingual synthetic data from GPT-4o.
>
> Firstly, for your major concern, our motivation is to leverage generative AI for more advanced embedding models and QA systems. Generative AI paves the way for representation learning that goes beyond contrastive learning. **Therefore, utilizing generative capabilities of language models for representation learning is itself a key contribution of our work.**
>
> In fact, **multilingual-E5 is contrastively pre-trained on billions of multilingual text pairs, and supervisedly fine-tuned on about 2 million high-quality multilingual data [5]**. The 500k data generated by GPT-3.5/4 is used for instruction tuning of the mE5-large-instruct model. Hence, these models have been fed with sufficient multilingual data. Otherwise, Microsoft can readily scale beyond the 2 million figure.
>
> Secondly, **we have supplemented the results on 15 monolingual or bilingual models in the appendix (see Table 9). The results confirm that, the improvements stem from our proposed method rather than multilingual data**.
>
> |Model|Param|FIGNEWS(English)||CRUD-RAG(Chinese)||
> |-|-|-|-|-|-|
> |||MRR@10|NDCG@10|MRR@10|NDCG@10|
> |**e5-large-v2**|-|73.2|77.2|-|-|
> ||$QAE_{hyb}$, $\alpha=0.3, \beta=1.0$|**78.7**|**82**|-|-|
> |**bge-large-en-v1.5**|-|66.4|71|-|-|
> ||$QAE_{hyb}$, $\alpha=0.15, \beta=1.5$|**74.3**|**78.2**|-|-|
> |**bge-large-zh-v1.5**|-|-|-|76.8|80|
> ||$QAE_{hyb}$, $\alpha=0.45, \beta=1.25$|-|-|**89**|**90.7**|
>
> In fact, **we adopted the cheapest GPT-4o-mini rather than GPT-4o, which is much weaker than GPT-3.5/4** and costs only 1/30th of the price [2]. Besides, 0.2B or 0.5B model is sufficient for question generation (See F3).
>
> For the E5 series [3], the multilingual-e5-large is currently the most popular E5 model, while the English-only e5-large-v2 was released in 2022. Table 3 in [3] reports, the multilingual-E5 is as strong as English-only models like bge-large-en-v1.5 and e5-large-v2 for English tasks [4].
>
> Finally, we admire classical benchmarks like BEIR, but they fail to fully reflect the capabilities of SOTA models in dynamic scenarios (See F4). Thanks!
>
> ### Response to W3
> > However, the hypothesis seems less connected to the method, and the introduction of overly complex concepts makes it less accessible.
>
> Firstly, we acknowledge the room for improvements on our presentation. **However, other reviewers confirmed the quality of our presentation overall with minor confusion.** E.g., Reviewer sfJJ mentioned the detailed description of fingerprints instead of the concept itself.
>
> We greatly appreciate your academic taste in simple and direct ideas. The main divergence in different opinions is the significance of our hypothesis. **Through the proof by contradiction / ablation, we clarify this hypothesis is not a trivial wrap, but of great importance.**
>
> Our hypothesis consists of two sub-hypotheses: the single-cluster hypothesis and the perpendicular hypothesis.
>
> - If the single-cluster hypothesis does not hold, the mean pooling estimation for the cluster center may **fail to converge or become meaningless**. (rationale)
> - If the perpendicular hypothesis is invalid or insignificant (i.e., the cluster center and the original document embedding are close), substituting the cluster center for the document embedding would **not result in improvements**. (effectiveness)
>
> Even a simple conclusion is supported by complexity. **Though our method is straightforward, the hypothesis underpins its rationale and effectiveness.**
>
> We hope these clarifications address your concerns and provide justification for reconsidering our work. If our response resolves your concerns, we kindly ask you to consider raising the rating of our work. Thank you very much for your time and efforts!
>
> [1] https://huggingface.co/datasets/microsoft/ms_marco
>
> [2] https://openai.com/api/pricing
>
> [3] Multilingual E5 Text Embeddings: A Technical Report
>
> [4] https://huggingface.co/BAAI/bge-large-en-v1.5
>
> [5] https://huggingface.co/intfloat/multilingual-e5-large

---

> > ### Comment · Reviewer_zDcY · 2024-12-03
> >
> > Thank you for your response. I have reservations regarding your rebuttal on W2 and W3, and it seems that W1 remains unresolved as the deadline approaches. Therefore, I will maintain my original rating.

---

### Official Review · Reviewer_QYtk · 2024-10-30

**Soundness:** 2
**Presentation:** 3
**Contribution:** 2
**Rating:** 5
**Confidence:** 5

**Summary:**

This paper proposes QAEncoder, a training-free method aimed at bridging the semantic gap between user queries and documents in dense retrieval systems for question answering (QA). The authors introduce the conical distribution hypothesis, suggesting that potential queries and documents form a cone-like structure in the embedding space. By estimating the mean of potential query embeddings, QAEncoder generates a surrogate embedding for each document to improve alignment with user queries. Additionally, the paper introduces document fingerprint strategies to maintain document distinguishability. Extensive experiments across multiple datasets, languages, and embedding models demonstrate that QAEncoder enhances retrieval performance without additional training or significant computational overhead.

**Strengths:**

1. The paper introduces a training-free method to improve query-document alignment, which can be valuable for systems where retraining is impractical. It seems to improve the performance of unsupervised models more, helping the cases that fewer fine-tuning data is available.
2. The authors conduct experiments across various datasets, languages, and embedding models, demonstrating the general applicability of QAEncoder.
3. QAEncoder is designed as a plug-and-play solution, which can be easily integrated into existing systems without significant modifications.

**Weaknesses:**

1. The performance gain over supervised models is marginal. In certain cases, it seems to worsen the scores on MSMARCO and HotpotQA (shown in the appendix).
2. If I understand correctly, the pseudo-queries are generated based on the corpus used for testing. This is equivalent to domain adaptation, so I think more domain adaptation baselines should be included, such as GPL, Combination of AdaLM and IDF, AugTriever.
3. A basic but reasonable baseline setting is also missing, which fine-tunes each model using the pairs of generated queries and documents. Even though it is fine-tuned and domain-adapted, it saves the cost for serving new documents.
4. Although QAEncoder doesn’t require training, generating multiple queries and embeddings per document could lead to computational strain. A closer look at the resources required would help readers understand if this approach scales well. Need an extra table to show the time of usage.

**Questions:**

1. Is there any statistics showing pseudo queries generated by GPT-4o-mini for each dataset (total number, number per doc, length, etc.)? If each doc comes with multiple generated queries, it can also be costly.
2. Why was the study not run on more comprehensive benchmarks like BEIR?
3.  Could the authors provide more detail on the computational resources and time required during indexing? Insight into this would clarify its applicability to large-scale datasets.

---

> ### Author Response · Authors · 2024-11-21
> **Author Response [1/2]**
>
> Dear Reviewer QYtk:
>
> Thank you for your thorough review and feedback. Below are responses to your comments and questions.
>
> ## Response to W1:
> > **(Performance Improvement)** The performance gain over supervised models is marginal. In certain cases, it seems to worsen the scores on MSMARCO and HotpotQA.
>
> We clarify the performance gain is generally significant over SOTA supervised models.
>
> On one hand, **the performance gain on classical datasets can be significant**. We acknowledge that the document-query gap of SOTA models has been alleviated on classical datasets. This is because the six classic datasets (MSMARCO, HotpotQA, ELI5, NQ, SQuAD, and TriviaQA) are frequently used for fine-tuning, i.e. supervised domain-adaption. For instance, BGE and E5 models have been fine-tuned on all of these datasets \[1,4\], which should reach performance limits on these datasets. However, they still suffer common issues in multi-task learning like **negative transfer and task imbalance** despite classical QA datasets being in similar domains. E.g., **E5 performs poorly on its fine-tuned dataset ELI5, and MRR increases from 39.0 to 46.4 with QAEncoder equipped.** The training-free characteristic of the QAEncoder provides more robustness than domain-adaption.
>
> On the other hand, **the performance gain on the latest datasets is consistently significant**.
> The open-domain retrieval performance on **rapidly and lifelong updated datasets** is of great interest, where in-domain adaptation is challenged and not effective anymore. Experiments on the latest datasets show the **significant and consistent improvements by QAEncoder on all tested models, datasets, and languages**. E.g., the MRR metric of gte-multilingual-base on the FIGNEWS(English) increases from 65.3 to 75.3 with QAEncoder equipped.
>
> ## Response to W2:
> > **(Domain Adaptation Baselines)** Generating pseudo-queries for corpus is equivalent to domain adaptation. More domain adaptation baselines should be included, such as GPL, CAI and AugTriever.
>
>
> Thank you for your insightful feedback. Domain adaption is helpful, especially unsupervised domain adaption methods. However, **QAEncoder differs from traditional domain adaption in many ways**.
>
> Firstly, both GPL and AugTriever use pseudo-queries for contrastive learning \[5,6\], while QAEncoder is training-free. Combination AdaLM and IDF (CAI) \[7\] is designed for sparse retrievers, and thus is also quite different.
>
> Secondly, QAEncoder enables **more sophisticated representations like multi-vector representation** with Gaussian Mixture Models, to capture more complex data distribution (future work discussed). This represents a significant departure from traditional domain adaptation techniques.
>
> Thirdly, the training-free nature of **QAEncoder circumvents issues like catastrophic forgetting, limited generalization and task imbalance in multi-domain even open-domain adaption** \[2\]. This feature is particularly **desirable and indispensable in rapidly and lifelong updated index**. Besides, as far as we know, existing domain-adaption methods mainly focus on a single domain.
>
> We still acknowledge the excellence of these related works. Discussion on domain-adaption methods like **GPL, CAI and AugTriever will be included in our revision**.
>
> ## Response to W3:
> > **(Domain Adaption Baselines)** Fine-tuning each model using the pairs of generated queries and documents should be a basic but reasonable baseline.
>
>
> We clarify that there are baselines with domain adaption, even supervised domain adaption.
>
> As responded in W1 and mentioned in our paper, SOTA models are extensively and supervisedly fine-tuned on classical datasets. **E.g., BGE and E5 models are fine-tuned on all tested classical datasets \[1,4\]. Hence, the results of SOTA embedding models on classical datasets provide the multi-domain adaption performance**, which not only indicates negative transfer or task imbalance issues of domain adaption, but also confirms the effectiveness of QAEncoder. We will make this conclusion more explicit in our revision.
>
> Finally, we appreciate the effectiveness of domain-adaption on single or closed domain. We will discuss these approaches and clarify the distinctions in the revision.
>
> ## Response to W4:
> > **(Efficiency & Cost-Effectiveness)** Generating multiple queries and embeddings per document could be costly. More information on cost and time usage is needed.
>
> Briefly, the computational overhead of QAEncoder is entirely acceptable. Please refer to Q1 and Q3 for details.

---

> ### Author Response · Authors · 2024-11-21
> **Author Response [2/2]**
>
> ## Response to Q1:
> > **(Pseudo Query Generation)** Is there any statistics showing pseudo queries generated by GPT-4o-mini for each dataset? If each doc comes with multiple generated queries, it can be costly.
>
> In our implementation, the **number of generated queries linearly depends on the document length**. For a chunk size of approximately 1024, **10 queries are typically enough**, each having an average length of approximately 6-15 tokens. As shown in Figure 6, for documents with a length greater than 150 words from MSMARCO datasets, the cluster center estimation with **10 queries exhibits a similarity score of 0.96 compared with 80 queries**. For MSMARCO (\~56 words/doc) and NQ (~78 words/doc), about 6 and 8 queries are generated per document respectively. Please see the response to Q3 for more cost details.
>
> ## Response to Q2:
> > **(BEIR Benchmark)** Why was the study not run on more comprehensive benchmarks like BEIR?
>
> Thank you for the suggestion. We admire BEIR as a comprehensive and well-established benchmark collection. While BEIR includes multiple datasets, **only three datasets are explicitly categorized under the focused QA task**, i.e. NQ, HotPotQA and FiQA-2018. We have adopted the two most prominent datasets, NQ and HotPotQA, for testing. Actually, **we utilized KILT \[11\] from Facebook for benchmark. KILT benchmark provides NQ, HotPotQA, TriviaQA and ELI5 datasets**, which are also well-established and all adopted in our research.
>
> ## Response to Q3:
> > **(Computational Cost and Time)** Could the authors provide more detail on the computational resources and time required during indexing? Insight into this would clarify its applicability to large-scale datasets.
>
> Briefly, QAEncoder is cost-effective and efficient.
>
> Firstly, the cost mainly comes from the query generation process. As we explained in Q1, **10 queries with an average length of 6-15 tokens is typically sufficient.**
>
> Secondly, **GPT-4o-mini is currently the cheapest generative model by OpenAI**, priced at \\$0.075 per million input tokens and \\$0.300 per million output tokens \[8\]. The most recent Qwen2.5-Turbo by Alibaba \[9\] is even 3.6 times cheaper than GPT-4o-mini. Hence, about **0.1 million documents can be processed within 1 dollar** via API call ($\frac{1\mathrm{M}\*3.6}{0.3*(10*10)} \approx 0.1\mathrm{M}$). The prices will continually decrease as AI and small language models develop.
>
> Thirdly, the initial query-centric work for sparse retrievers, DocT5Query \[10\], demonstrates **T5-base with 0.2B parameters is sufficient for query generation, while no improvements are gained with larger models.** We also confirmed Qwen2.5-0.5B-Instruct as a good query generator in our business implementation. We choose GPT-4o-mini just given its **out-of-the-box and comprehensively multilingual support for academic research**.
>
> Finally, in our data flow settings (batch size=1), 2\~3 documents can be processed per second with Qwen2.5-0.5B-Instruct + vLLM + BF16 on a single NVIDIA A100 80GB\[3\]. For batch processing with higher GPU utility, DocT5Query reports sampling 5 queries per document for **8.8M MSMARCO documents requires approximately 40 hours on a single Google TPU v3, costing only \$96 USD** (40 hours × \$2.40 USD/hour) in 2019 \[10\].
>
> Hence, QAEncoder is indeed **acceptable and cost-effective**.
>
> Thanks again for the detailed comments. Your feedback is valuable for improving our paper. Let us know if there are still any concerns.
>
> ## Reference
> [1] https://huggingface.co/datasets/Shitao/bge-m3-data
>
> [2] Domain adaptation and multi-domain adaptation for neural machine translation: A survey
>
> [3] https://qwen.readthedocs.io/en/latest/benchmark/speed_benchmark.html
>
> [4] https://huggingface.co/intfloat/multilingual-e5-large
>
> [5] GPL: Generative pseudo labeling for unsupervised domain adaptation of dense retrieval
>
> [6] Augtriever: Unsupervised dense retrieval by scalable data augmentation
>
> [7] Unsupervised domain adaptation for sparse retrieval by filling vocabulary and word frequency gaps
>
> [8] https://openai.com/api/pricing
>
> [9] https://qwen2.org/qwen2-5-turbo
>
> [10] From doc2query to docTTTTTquery
>
> [11] KILT: a benchmark for knowledge intensive language tasks

---

> ### Author Response · Authors · 2024-11-24
> **Official Comment by Authors**
>
> Dear reviewer QYtk:
>
> Thanks again for your valuable comments. We hope that our response could address your concerns. Please do not hesitate to ask if you have any questions about our manuscript or response.

---

### Official Review · Reviewer_sfJJ · 2024-11-04

**Soundness:** 3
**Presentation:** 3
**Contribution:** 2
**Rating:** 5
**Confidence:** 4

**Summary:**

The paper introduces QAEncoder, a training-free method designed to enhance alignment in RAG systems by addressing the document-query gap. The approach is based on the conical distribution hypothesis, which suggests that potential queries and documents form a cone-like structure in embedding space. QAEncoder estimates the cluster center of potential queries as a surrogate for document embeddings, improving alignment without increasing storage or retrieval latency. The method integrates seamlessly with existing RAG architectures and includes strategies like document fingerprinting to maintain document distinguishability. Extensive experiments demonstrate QAEncoder's effectiveness across multiple languages and datasets.

**Strengths:**

1. Excellent architecture and visualization figures.
2. The main experiment includes many different models and hyperparameters.
3. A series of ablation experiments were conducted.
4. The proposed method is effective and allows pre-computation of the (without query) embedding part, accelerating the inference process.

**Weaknesses:**

1. The descriptions of several QAE fingerprints are somewhat confusing.
2. There are relatively many hyperparameters, and the model's performance varies significantly across different scenarios and hyperparameter settings.

**Questions:**

1. Why not combine $QAE_{emb}$ and $QAE_{txt}$?
2. How can you find the most suitable hyperparameters in different scenarios? Is it cumbersome to rely on repeated searches? Do you have any insights on simpler methods?
3. Maybe the QAEncoder framework can be used in retrieval model's training?

---

> ### Author Response · Authors · 2024-11-21
> **Author Response**
>
> Dear Reviewer sfJJ:
>
> Thank you for your careful review and insightful feedback. Below are our responses to address your concerns.
>
> ## Response to W1:
> > **(Document Fingerprint Strategies)** The descriptions of several fingerprints are somewhat confusing.
>
> QAEncoder's initial form, $QAE_{base}$, faces the document distinguishability issue, as all representations become query-like. Hence, **documents' characteristic information, intuitively termed fingerprints, are integrated for document uniqueness.**
>
> $QAE_{emb}$ and $QAE_{txt}$ extend $QAE_{base}$ with fingerprints in the embedding and textual spaces respectively. As an exploratory attempt, $QAE_{hyb}$ is proposed to combine $QAE_{emb}$ and $QAE_{txt}$. Please refer to Q1 for more consideration on the combination. Let us know if any part requires more clarification.
>
> ## Response to W2:
> > **(Hyperparameter Robustness Analysis)** There are relatively many hyperparameters, and the performance varies significantly across different scenarios and hyperparameter settings.
>
> First, there are **only two hyperparameters**, $\alpha$ for linear interpolation and $\beta$ for length control.
>
> Then, ablation studies shows that, $QAE_{hyb}$ and $QAE_{emb}$ generally outperform $QAE_{txt}$, while the best performance of $QAE_{emb}$ and $QAE_{hyb}$ is notably close. Hence, **the search on $QAE_{emb}$ is sufficient for competitive performance.** Please refer to Q2 for more HP search suggestions. Thanks.
>
> ## Response to Q1:
> > **(Combination of $QAE_{emb}$ and $QAE_{txt}$)** Why not combine $QAE_{emb}$ and $QAE_{txt}$?
>
> We acknowledge the direct combination of $QAE_{emb}$ and $QAE_{txt}$ is intuitive and could lead to potential improvement. However, **this combination introduces three hyperparameters with significantly enlarged search space but limited improvement.**
>
> For instance, $QAE_{hyb}(d) =  (1 - \alpha) \cdot QAE_{txt}(d) + \alpha \cdot QAE_{base}(d)$. $QAE_{hyb}$ generalizes $QAE_{emb}$ with one more hyperparameter but minor performance improvement. The combination of $QAE_{emb}$ and $QAE_{txt}$ generalizes $QAE_{hyb}$ once again. Similarly, we conducted trials with no significant improvement found.
>
> Briefly, this is a **trade-off** between improvement and complexity.
>
> ## Response to Q2:
> > **(Hyperparameter Search Suggestion)** To find the most suitable hyperparameters in different scenarios, do you have simpler methods than repeated searches?
>
> As responded in W2, $QAE_{emb}$ maintains competitive performance with a single hyperparameter. Hence, **$QAE_{emb}$ is recommended for accelerating the HP search**.
>
> Firstly, we believe that the optimal hyperparameters are primarily influenced by the inherent characteristics of the embedding model, i.e. the geometric property of embedding space. Therefore, **a one-turn search should be sufficient for a given embedding model.** That's why we optimize hyperparameters simultaneously across multiple datasets.
>
> Secondly, **the one-turn search can also be accelerated**. As Figure 4 shows, the performance of $QAE_{emb}$ initially rises and then falls as $\alpha$ increases, peaking between 0.3 and 0.6. This concave property enables **ternary search [1] with logarithmic trails** rather than brute-force search.
>
> Finally, the property of datasets also slightly influences the optimal hyperparameters. Specifically, the optimal $\alpha$ for classical datasets is marginally lower than that for latest datasets (refer to Tables 5 and 6 for details). Therefore, **selecting the optimal $\alpha$ based on classical datasets represents a cautious and robust strategy**, ensuring consistent improvement across both classical and latest datasets.
>
> Hope this response provides some insights for HP search.
>
> ## Response to Q3:
> > **(Retrieval Model Training)** Maybe the QAEncoder framework can be used in retrieval model's training?
>
> Yes, there are many possibilities for QAEncoder in model training. E.g., **new regularization terms with the conical distribution hypothesis as prior knowledge** in contrastive learning.
>
> More directly, we have tried using a three-layer MLP with MSE loss to **transform the original document embedding into the cluster center $QAE_{base}$. That is, the one-step conversion without query generation**. Trained on 40,000 samples and tested on 10,000, our model achieved average **cosine similarities of 0.97 (training) and 0.92 (testing)**, between predicted embeddings and actual cluster centers. After scaling with $\alpha$, the approximation of $QAE_{base}$ is adequate.
>
> Though this preliminary result highlights the potential of QAEncoder in model training. We must emphasize, the training-free nature of QAEncoder provides it with a unique advantage in robust and lifelong generalization than training-based methods.
>
> Finally, thanks again for the valuable suggestions. Hope our responses address your concerns. Feel free to contact us if you have more questions or suggestions.
>
> ## Reference
>
> [1] https://en.wikipedia.org/wiki/Ternary_search

---

> > ### Comment · Reviewer_sfJJ · 2024-11-25
> >
> > Thank you for your responses. I have carefully reviewed them. One suggestion I would like to offer is the following:
> >
> > > This concave property enables ternary search [1] with logarithmic trails rather than brute-force search.
> >
> > It may be worthwhile to conduct a theoretical analysis of this property. Additionally, your conical distribution hypothesis could potentially be evaluated in this context.
> > However, since this observation stems from your empirical discoveries, it may be premature to definitively conclude that the property is "concave."

---

> > > ### Author Response · Authors · 2024-11-26
> > > **Author Response (2nd Round)**
> > >
> > > Dear Reviewer sfJJ,
> > >
> > > Thank you very much for your constructive feedback.
> > > > However, since this observation stems from your empirical discoveries, it may be premature to definitively conclude that the property is "concave."
> > >
> > > We acknowledge that the mentioned "concave property" in our response lacks mathematical rigor. However, we emphasize that, regarding the hyperparameter $\alpha$, the performance curve of $QAE_{emb}$ consistently exhibits a unimodal phenomenon in practice. Therefore, it still enables ternary search.
> > >
> > > As a preliminary theoretical analysis, this is attributed to the interplay between mitigating the document-query gap and the side effect of document indistinguishability, which together determine the optimal balance point.
> > >
> > > As you noted, exploring more advanced query-to-query matching methods beyond mean pooling is worthwhile. We believe that our approach could inspire more sophisticated matching algorithms that will alleviate the aforementioned side effect and achieve greater performance improvements.
> > >
> > > Thank you once again for your valuable insights. Please feel free to reach out if you have any questions regarding our manuscript or our response.

---

> ### Author Response · Authors · 2024-11-24
> **Official Comment by Authors**
>
> Dear reviewer sfJJ:
>
> Thanks again for your valuable comments. We hope that our response could address your concerns. Please do not hesitate to ask if you have any questions about our manuscript or response.

---

### Official Review · Reviewer_coVe · 2024-11-06

**Soundness:** 3
**Presentation:** 3
**Contribution:** 3
**Rating:** 6
**Confidence:** 4

**Summary:**

The paper proposes a simple but novel document encoding method based on a set of these corresponding (automatically generated) queries, QAEncoder, which generates a set of queries first given a document according to 5W1H, and considers the mean embedding of queries as a document embedding. The experiment results show that the retrieval performance is substantially improved.

**Strengths:**

1.	This paper proposes a simple but novel document embedding method to reduce the gap b/w query-document matching, based on a set of queries automatically generated for a given document. It is well-motivated, and it is interesting to see that the simple idea has not been suggested to my best of knowledge, while the work is indeed novel.
2.	The experiments show that the proposed QAencoding leads to improvements over the baseline dense retrieval, while more baselines need to be included.

**Weaknesses:**

1.	The paper assumes Hypothesis, but it is not clear how this hypothesis is connected to the final form of QAencoder. The hypothesis says “the document embedding lies in the perpendicular line intersecting the cluster center”, but it is not clear whether the final embedding vector (Eq. 3 and other equations) fulfills this condition, where lambda is positive. Specially, it seems that the conical assumption hypothesis is not important on this paper, because it doesn’t have a serious role to my understanding. All the methods could be presented, without the conical assumption hypothesis. More explanation is desirable, because this part makes the paper less-readable.
2.	More baselines need to be provided. In particular, different from the proposed direction, the reverse direction (i.e. query2document style) also needs to be compared.
3.	Only mean pooling of query embeddings is used. However, taking the mean pooling may make individual query semantics blurred. Advanced pooling or matching methods for query-to-query may be examined and discussed.

**Questions:**

Please see weaknesses.

---

> ### Author Response · Authors · 2024-11-21
> **Author Response**
>
> Dear Reviewer coVe:
>
> Thank you for your constructive feedback and efforts on our research. We greatly appreciate the time and effort you have invested. Let us address your questions one by one.
>
> ## Response to W1:
> > **(Hypothesis's Significance)**
> The hypothesis posits that document embeddings intersect the cluster center perpendicularly, yet it's unclear if this holds for the final embeddings. The role of the conical distribution hypothesis appears minor, as the methods seem presentable without it.
>
> **Briefly, the conical distribution hypothesis is targeted at naive embedding models.** This hypothesis not only motivates the design of QAEncoder largely, but also serves as the foundation for all the methods presented.
>
> Firstly, this hypothesis is discovered on naive embedding models like BGE and E5, which have not been boosted by QAEncoder. A more precise expression would be: **"the original document embedding lies in the perpendicular line intersecting the cluster center"**. This hypothesis unveils the limitations of the original document embedding.
>
> Secondly, this **hypothesis is significant as it captures the geometric relationship** between documents and potential queries in the embedding space of the naive embedding model. It ensures the rationality and effectiveness of the $QAE_{base}$ and final form design.
>
> - The initial form, $QAE_{base}$, is directly derived by this hypothesis. **Under this hypothesis, $QAE_{base}$, which estimates queries' cluster center as the document surrogate, theoretically aligns best with potential queries.**
>
> - To enhance document distinguishability, $QAE_{base}$ is further combined with document fingerprints, which leads to the final forms $QAE_{emb}$, $QAE_{txt}$ and $QAE_{hyb}$.
>
>
> ## Response to W2:
> > **(Baselines)** More baselines are needed. In particular, different from the proposed direction, the reverse direction (i.e. query2document style) needs to be compared.
>
> Briefly, we clarify that the reverse direction (i.e. query2document style) like HyDE is compared, and several baselines are also included.
>
> Firstly, **the well-known document-centric model HyDE, which generates pseudo documents for each query is tested**. As shown in Tables 4 and 8, retrieval performance of HyDE exhibits a **significant decline, due to severe hallucination issues** with out-of-domain new knowledge.
>
> | Model                 | Param    | FIGNEWS(English) |          | FIGNEWS(Arabic) |          | CRUD-RAG(Chinese) |          |
> | --------------------- | -------- | ---------------- | -------- | --------------- | -------- | ----------------- | -------- |
> |                       |          | MRR@10           | NDCG@10  | MRR@10          | NDCG@10  | MRR@10            | NDCG@10  |
> | mcontriever           | -        | **32.9**         | **36.7** | **40.3**        | **44.7** | **71.8**          | **76.2** |
> |                       | HyDE,n=8 | 25               | 27.9     | 35.7            | 41.9     | 70.9              | 74.7     |
> | multilingual-e5-large | -        | **73.9**         | **77.8** | **76.7**        | **80.2** | **85.8**          | **88.3** |
> |                       | HyDE,n=8 | 63.6             | 68.3     | 68.3            | 74.1     | 81.6              | 84.1     |
>
> Secondly, **extensive experiments are conducted in comparison with 14 embedding models**. Especially, as SOTA models are extensively fine-tuned on classical datasets, the results of SOTA models on classical datasets demonstrate the capabilities of domain adaption methods. Table 1 shows multi-domain adaption suffers negative transfer and task imbalance issues, while the training-free nature of QAEncoder avoids these.
>
> Thirdly, we also discussed the relationship between QAEncoder and training-based methods. Table 4 shows, **QAEncoder help training-based methods further enhance performance**. E.g., multilingual-e5-large-instruct increases 8.8 MRR points on FIGNEWS(English) with QAEncoder, which is instruction-tuned before.
>
> To sum up, our work presents the comparative analysis of various baselines.
>
> ## Response to W3:
> > **(More Sophisticated Representations)** The use of mean pooling for query embeddings can blur individual query semantics, more advanced pooling or matching methods may be discussed.
>
> Thank you for your insightful and interesting suggestion. **Besides the document fingerprint strategies we examined, one possibility can be multi-vector representation.**
>
> Under the single-cluster sub-hypothesis, our work provides mean pooling as a simple yet effective approach. We acknowledge the single cluster may not fully capture the complexity of query distributions. Under more complicated multi-cluster hypothesis, **multi-vector representation based on GMM can be devised for more nuanced representation**. We orient the initial QAEncoder as a simple yet effective work, and leave these potential **in the future work**. Thanks again!
>
> Hope our responses address your concerns. Let us know if there are still any concerns!

---

> ### Author Response · Authors · 2024-11-24
> **Official Comment by Authors**
>
> Dear reviewer coVe：
>
> Thanks again for your valuable comments. We hope that our response could address your concerns. Please do not hesitate to ask if you have any questions about our manuscript or response.

---

> ### Author Response · Authors · 2024-11-25
> **More Author Response**
>
> Dear reviewer coVe：
>
> Thanks again for your valuable comments. We would like to provide a more structured and clear explanation of our hypothesis's significance.
>
> ### Response to W1:
> > **(Hypothesis's Significance)** The role of the conical distribution hypothesis appears minor, as the methods seem presentable without it.
>
> We elucidate its critical role through **the proof by contradiction / ablation**.
>
> Our hypothesis consists of two sub-hypotheses: the single-cluster hypothesis and the perpendicular hypothesis.
>
> - If the single-cluster hypothesis does not hold, the mean pooling estimation for the cluster center may **fail to converge or become meaningless**. (rationale)
> - If the perpendicular hypothesis is invalid or insignificant (i.e., the cluster center and the original document embedding are close), substituting the cluster center for the document embedding would **not result in improvements**. (effectiveness)
>
> Even a simple conclusion is supported by complexity. **Though our method is straightforward, the hypothesis underpins its rationale and effectiveness.**
>
> We hope that our response could address your concerns. Please do not hesitate to ask if you have any questions about our manuscript or response.

---

### Author Response · Authors · 2024-11-24
**General Response**

We sincerely thank all the reviewers for their valuable and constructive comments. The core contributions, general responses to common concerns, and paper supplements are as follows:

## Core Contributions
- **Methodological Innovations.** We pioneer to bridge the document-query gap in dense retrievers from the query-centric perspective. QAEncoder mainly estimates the expectation of potential queries in the embedding space as a robust surrogate for the document embedding in a training-free manner. It not only avoids extra index storage, retrieval latency, catastrophic forgetting and hallucination, but also guarantees diversified query handling and robust generalization.
- **Theoretical Discovery.** We formulate the conical distribution hypothesis on the geometry structure in embedding space, providing a theoretical foundation for the alignment of document and query embeddings, which is validated through extensive empirical analysis.
- **Practical Applications.** QAEncoder is designed as a truly plug-and-play solution, seamlessly integrating with existing RAG architectures and training-based methods.

## General Responses
There are two common concerns, (1) the cost-effectiveness of QAEncoder, and (2) the performance improvement over SOTA baselines.
- ### **Cost-Effectiveness**

Briefly, QAEncoder is cost-effective and efficient.

The cost mainly comes from the query generation process. For query generation, 10 queries are typically sufficient for ordinary documents. Therefore, about **0.1 million documents can be indexed within 1 dollar via API call**. And the prices will continually decrease as AI and small language models develop.

For local deployment, as reported, T5-base with 0.2B parameters is sufficient for query generation. The processing of **8.8 million MSMARCO documents requires approximately 40 hours on a single Google TPU v3, costing only \$96 USD** in 2019. In our data flow settings (batch size=1), 2~3 documents can be processed per second.

In our work, we choose GPT-4o-mini given its **out-of-the-box and comprehensively multilingual support for academic research**. For more details, please refer to Fact-3 (Cost-Effectiveness) in our response to Reviewer zDcY.

- ### **Performance Improvement over SOTA Baselines**

Briefly, the performance improvement over SOTA baselines is significant. We employed several SOTA baselines extensively fine-tuned on well-established datasets.

On the one hand, SOTA models on classical datasets (domain adaptation setting) suffer from common issues in multi-task learning like negative transfer and task imbalance. E.g., **E5 performs poorly on its fine-tuned dataset ELI5, and MRR increases from 39.0 to 46.4 with QAEncoder equipped**.

On the other hand, SOTA models on latest datasets (rapid and lifelong knowledge update setting), QAEncoder **significantly and consistently  improves on all tested models, datasets and languages**.

For more details, please refer to Fact-4 (Experiment Design and Results) in our response to Reviewer zDcY.

## Paper Supplements and Modifications
- The discussion on domain adaptation methods (e.g., GPL, CAI, and AugTriever) in related works section [Reviewer QYtk, zDcY].
- Performance of DPR as a baseline method [Reviewer zDcY], with results presented in Table 1 and Table 5.
- Comprehensive results for key metrics (MRR, MAP, NDCG) with top-k = 10 in the Appendix [Reviewer zDcY], presented in Tables 10, 11, 12, and 13.
- Further experiments on SOTA monolingual and bilingual models for stronger baselines [Reviewer zDcY], presented in Table 9.
- Hyperparameter search and selection suggestions [Reviewer sfJJ], presented in the appendix.

--------------


Finally, we are very grateful for the reviewers' high evaluation of our work, such as:

**Reviewer coVe:**
> It is well-motivated, and it is interesting to see that the simple idea has not been suggested, while the work is indeed novel.

**Reviewer sfJJ:**
> Excellent architecture and visualization figures.
>
> The proposed method is effective and allows pre-computation of the embedding part, accelerating the inference process.

**Reviewer QYtk:**
> The authors conduct experiments across various datasets, languages, and embedding models, demonstrating the general applicability of QAEncoder.

**Reviewer zDcY:**
> Some of their findings are interesting and may provide insights into more effective techniques for representation augmentation.

We again thank all the reviewers, and look forward to further discussions and responses from the reviewers!

---

### Note · Authors · 2024-12-10

**Comment:**

After discussion, the authors have unanimously decided to withdraw the manuscript.
We thank the reviewers and the AC for their time.

**Withdrawal Confirmation:**

I have read and agree with the venue's withdrawal policy on behalf of myself and my co-authors.